# Vesicular glutamate transporters are H+-anion exchangers that operate at variable stoichiometry

Bettina Kolen [1,3], Bart Borghans [1,3], Daniel Kortzak [1], Victor Lugo[1], Cora Hannack [1], Raul E. Guzman [1], Ghanim Ullah [2] & Christoph Fahlke [1] ✉

Vesicular glutamate transporters accumulate glutamate in synaptic vesicles, where they also function as a major Cl⁻ efflux pathway. Here we combine heterologous expression and cellular electrophysiology with mathematical modeling to understand the mechanisms underlying this dual function of rat VGLUT1. When glutamate is the main cytoplasmic anion, VGLUT1 functions as H+-glutamate exchanger, with a transport rate of around 600 s⁻¹ at −160 mV. Transport of other large anions, including aspartate, is not stoichiometrically coupled to H+ transport, and Cl⁻ permeates VGLUT1 through an aqueous anion channel with unitary transport rates of $1.5 \times 10^5$ s⁻¹ at −160 mV. Mathematical modeling reveals that H+ coupling is sufficient for selective glutamate accumulation in model vesicles and that VGLUT Cl⁻ channel function increases the transport efficiency by accelerating glutamate accumulation and reducing ATP-driven H+ transport. In summary, we provide evidence that VGLUT1 functions as H+-glutamate exchanger that is partially or fully uncoupled by other anions.

Synaptic vesicles accumulate neurotransmitters via secondary active transporters that harness the electrochemical H+ gradient across the vesicular membrane. In the mammalian central nervous systems, three vesicular glutamate transporters (VGLUT1, VGLUT2 and VGLUT3) are responsible for filling the synaptic vesicles of excitatory neurons with glutamate[1–4]. VGLUTs are members of the SLC17 family, together with inorganic phosphate transporters, the lysosomal H+/sialic acid cotransporter sialin, and the vesicular nucleotide transporter VNUT[1]. Previous studies with vesicle preparations, with purified and reconstituted transporters or with a combination of heterologous expression and cellular electrophysiology demonstrated that VGLUTs are not only glutamate transporters, but also represent the main Cl⁻ transport pathway in synaptic vesicles[5–7]. VGLUTs were additionally shown to function as K+/H+ exchangers in the synaptic vesicle, in the presence as well as in the absence of glutamate[3,8,9], and to import phosphate from the extracellular space when residing in the plasma membrane[10].

Here, we combine heterologous expression of plasma membrane-targeted mutant VGLUT1 and whole-cell patch clamping with mathematical modeling to describe the mechanistic basis of the glutamate transport and the anion channel functions and their role in selective glutamate transport into synaptic vesicles. We find that VGLUT1 functions as H+-coupled anion exchanger that varies its transport stoichiometry to select between different substrates and to permit Cl⁻ efflux during glutamate accumulation.

## Results

### VGLUT1 functions as proton- and voltage-activated anion channel

VGLUT1 localizes to intracellular membranes of transfected cells, but alanine substitutions in N- and C-terminal dileucine-like motifs (EE6,7 AA/L11A/EE505,506AA/F510A/V511A VGLUT1, hereafter referred to as VGLUT1_PM) resulted in predominant surface membrane insertion

[1]Institute of Biological Information Processing, Molekular- und Zellphysiologie (IBI-1), Forschungszentrum Jülich, 52428 Jülich, Germany. [2]Department of Physics, University of South Florida, Tampa, FL 33620, USA. [3]These authors contributed equally: Bettina Kolen, Bart Borghans. ✉e-mail: c.fahlke@fz-juelich.de

in HEK293T cells (Supplementary Fig. 1)[6]. To study VGLUT1$_{PM}$-mediated Cl⁻ transport without the main transport substrate glutamate in whole-cell patch clamp recordings cells were dialyzed with a choline chloride-based internal solution at pH 7.4, and perfused with choline chloride-based external solutions of variable pH. At pH 6.5 and below, hyperpolarizing voltage steps elicited current activation, whereas currents were absent at positive potentials (Fig. 1a). Under identical ionic conditions, currents in non-transfected HEK293T cells or HEK293T cells expressing a non-functional mutant, H191K-H426K-D428Q VGLUT1$_{PM}$, were negligible (Supplementary Fig. 2a, b). Endogenous anion currents are not pH-dependent, and VGLUT1$_{PM}$ currents activated by acidic pH 5.5 can therefore be corrected for background currents by subtracting current amplitudes measured at neutral pH values at the same cell (Supplementary Fig. 2c, d). WT VGLUT1 (without improved plasma membrane targeting) currents were much smaller (at −160 mV, pH$_o$ = 5.5; 47 ± 27 pA, $n$ = 10 cells) than VGLUT1$_{PM}$ currents (1548 ± 332 pA, $n$ = 36 cells), but above currents in untransfected cells (22 ± 13 pA, $n$ = 10 cells, two-tailed $t$-test, $p$ = 0.002). In agreement with earlier reports[6], VGLUT1$_{PM}$ Cl⁻ currents are activated by external Cl⁻ and Br⁻, but not by I⁻ (Supplementary Fig. 3).

We expressed VGLUT1$_{PM}$ as an eGFP fusion protein and quantified the number of transporters by measuring the whole-cell fluorescence of every cell[11,12]. Figure 1b shows such analysis for cells expressing VGLUT1$_{PM}$ and dialyzed with Cl⁻, NO$_3$⁻, Br⁻, or I⁻-based internal solutions. For WT VGLUT1$_{PM}$, there is a strict monotonic relation between currents at −160 mV and fluorescences of separate cells under activating acidic conditions (pH 5.5, Fig. 1b). Currents at pH 7.4 are independent of VGLUT1$_{PM}$ expression levels (small symbols, Fig. 1b). We used whole-cell fluorescence to normalize macroscopic current

amplitudes for the number of expressed transporters and observed the largest normalized mean current amplitudes for NO$_3$⁻, followed by I⁻, Br⁻ and Cl⁻ (Fig. 1c). Figure 1d depicts the pH dependence of VGLUT1$_{PM}$ currents carried by two internal anions. Fits with Michaelis−Menten relationships provide a pK$_M$ = 5.5 ± 0.6 ($n$ = 27) for cells dialyzed with internal Cl⁻ or pK$_M$ = 6.1 ± 0.02 ($n$ = 12 cells, two-tailed $t$-test, $p$ = 0.00182) for cells dialyzed with internal NO$_3$⁻ (Fig. 1d).

We used noise analysis to describe the anion channel function of VGLUT1$_{PM}$ in more detail[13–16]. We first determined the power spectra from macroscopic anion currents using fast Fourier transformation[17] after isolating VGLUT1$_{PM}$ anion current noise by subtracting the spectral densities obtained at pH 7.4 from those obtained at pH 5.5 (Fig. 2a). VGLUT1$_{PM}$ anion current noise spectra fitted well with a sum of two Lorentzian components and a term describing pink noise. Corner frequencies $f$ of the Lorentzian components ($f_{C1}$ = 1456 Hz; $f_{C2}$ = 265 Hz) predict very fast opening and closing time constants. Experimental spectral densities exceeded the upper limit for the spectral density of uniporter-mediated shot noise (S = 4.4 × 10⁻²⁹ A²/Hz) obtained by Schottky's theorem (S(f) = 2Iq, with I = 273 ± 120 pA ($n$ = 13) as the mean macroscopic current in the experiments used to determine the power spectra and the elementary charge as the net transported charge per shot[17]). Non-transfected HEK293T cells had lower current variances and power spectra resembling 1/$f$ noise under the same conditions. These results demonstrate that VGLUT1 anion current-associated noise arises from the random opening and closing of individual channels.

We modified VGLUT1$_{PM}$ by variation of the external pH and measured stationary current noise at the end of 100−500 ms voltage steps to −160 mV (Fig. 2b). Current noise generated by channels that switch between active and inactive conformations depends on the

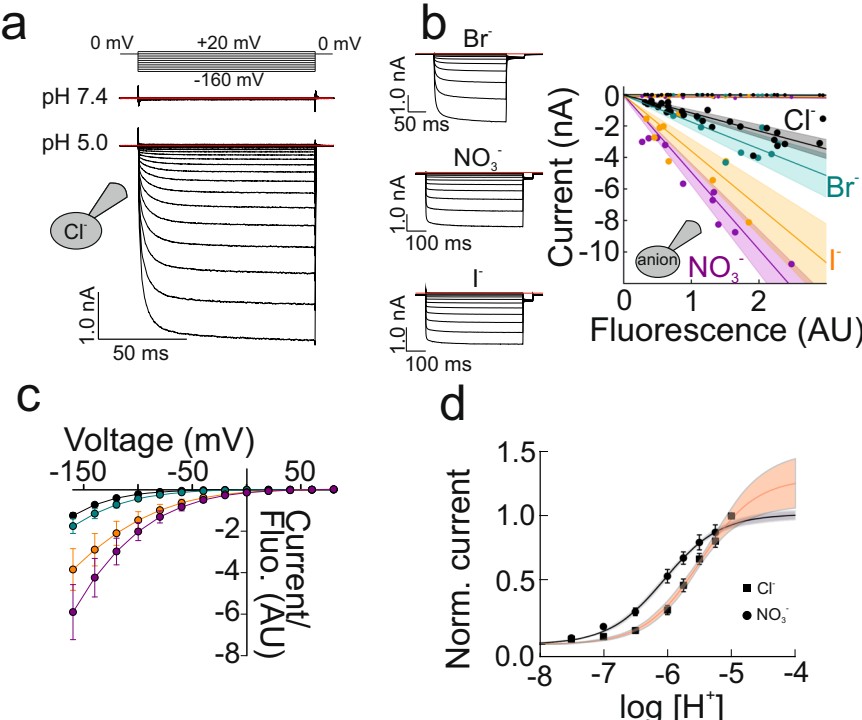

**Fig. 1 | VGLUT$_{PM}$ anion currents are pH- and voltage-dependent.**
**a** Representative recordings from a transfected cell with Cl⁻ as the internal anion perfused with Cl⁻-based solutions. No leakage subtraction was performed, and red lines represent zero current. **b** Plot of uncorrected whole-cell current amplitudes at the end of 500 ms voltage steps to −160 mV versus whole-cell fluorescences. Lines and shaded areas show the mean and 95% CI from linear fits. Measurements at pH 5.5 are given by large symbols, measurements at 7.4 by small symbols. Insets on the left show representative current responses to voltage steps between −160 and

+80 mV at external pH 5.5 and 40 mM external Cl⁻ with indicated internal anions. **c** Voltage dependence of background corrected whole-cell currents normalized to whole-cell fluorescence (means ± CI, $n$ = 10 cells for each internal anion). **d** External pH dependence of maximum current amplitudes at −160 mV (means ± CI, $n$ = 27 cells with internal Cl⁻, $n$ = 12 cells with internal NO$_3$⁻). Lines and shaded area depict mean and 95% CI from fits with Michaelis-Menten relationships. Source data are provided in the Source Data file.

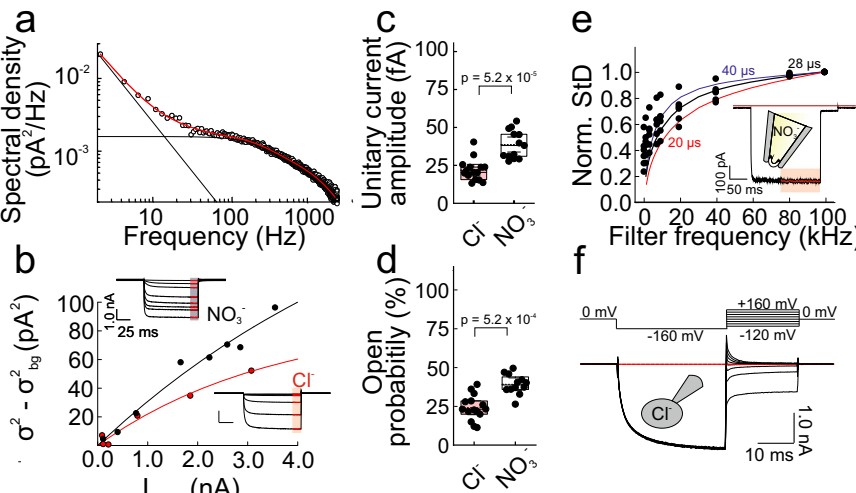

**Fig. 2 | VGLUT1$_{PM}$ functions as voltage-gated anion channels. a** Averaged power spectrum from 13 cells expressing VGLUT1$_{PM}$ with internal Cl⁻ at pH 5.5 after subtraction of background currents. The red line represents the fit with a second-order Lorentzian function in addition to pink noise ($S_1 = 9.2 \times 10^{-28}$ A²/Hz; $S_2 = 6.2 \times 10^{-28}$ A²/Hz, a = 1.36), and black lines the individual components of the fit.
**b** Representative current−variance plots of stationary noise analysis at −160 mV with internal Cl⁻ (red) or NO$_3^-$ (black). **c, d** Unitary current amplitudes (**c**) and absolute open probabilities (**d**) at −160 mV for cells dialyzed with internal Cl⁻ (red, $n = 15$ cells) or NO$_3^-$ (black, $n = 13$ cells). *P*-values from two-tailed *t*-test. Box plots show single data as symbols, mean values as dashed lines, medians as solid lines,

upper and lower quartiles as box borders and 95% CI as whiskers. **e** Changes in standard deviation (STD) of experimental current recordings upon filtering with Butterworth filters of different low-pass filter frequencies (symbols, $n = 6$ patches) and of simulated unitary current recordings with indicated open times (lines). Currents were obtained from excised outside-out patches with NO$_3^-$ as anion in the pipette solution at −160 mV; the inset depicts a representative recording.
**f** Representative responses of whole-cell currents with internal Cl⁻ to voltage steps consisting of an activating prepulse to −160 mV followed by variable test pulses. Source data are provided in the Source Data file.

unitary current amplitude i, the open probability p, and the number of channels N[18]:

$$\sigma^2 - \sigma_{bg}^2 = Np(i - ip)^2 + N(1-p)(0-ip)^2 = Ni^2 p(1-p) \quad (1)$$

$$\frac{\sigma^2 - \sigma_{bg}^2}{I} = i(1-p) \quad (2)$$

The variance by current ratio ($\frac{\sigma^2 - \sigma_{bg}^2}{I}$) rose with increasing external pH, demonstrating that external acidification increases the open probability rather than modifying the unitary current amplitude (Supplementary Fig. 4). Assuming constant unitary current amplitudes, the current noise is expected to change in a parabolic fashion with the mean current amplitude I[13,18]:

$$\sigma^2 - \sigma_{bg}^2 = iI - \frac{I^2}{N} \quad (3)$$

Figure 2b shows representative plots of current noise versus mean current amplitude for Cl⁻-based or NO$_3^-$-based internal solutions, fitted with Eq. 3. Figure 2c provides unitary current amplitudes from 15 cells with internal Cl⁻ ($24 \pm 1.1$ fA) and 13 cells with internal NO$_3^-$ ($39 \pm 1.2$ fA, two-tailed *t*-test, $p = 5.21 \cdot 10^{-5}$). These values correspond to transport rates of $1.5 \times 10^5 s^{-1}$ (intracellular Cl⁻) and $2.4 \times 10^5 s^{-1}$ (intracellular NO$_3^-$), which largely exceed the transport rates of known transporters, thus demonstrating that anion conduction occurs via diffusion along an aqueous conduction pathway[19,20]. Noise analysis also provided the number of channels per cell, and these values were used to calculate the absolute open probabilities (Fig. 2d). For these, we obtained lower values for Cl⁻ ($0.24 \pm 0.01$) than for NO$_3^-$ ($0.39 \pm 0.01$, two-tailed *t*-test, $p = 0.00052$).

Since filtering in the range of the single event duration modifies apparent unitary current amplitudes, the duration of channel openings can be estimated by quantifying the consequences of low pass filtering

on the current noise[15,21]. Figure 2e shows the dependence of normalized standard deviations of experimentally determined currents, which are a measure of the unitary current amplitude ($SD = i\sqrt{Np(1-p)}$)[15,21], on the filter frequency. Solid lines represent the effect of filtering on normalized standard deviations of simulated single channel currents with one open and one closed state[15] and with open times of 20, 28, and 40 μs. We conclude that VGLUT1 functions as Cl⁻ or NO$_3^-$ channel with very short open states.

VGLUT1 can thus function as Cl⁻ channel that opens and closes upon changes in voltages. A physiologically important consequence of VGLUT1 channel gating is its pronounced inward rectification (Fig. 1a, c). After an activating prepulse to −160 mV, voltage steps toward positive potentials elicited instantaneous currents that deactivated on a monoexponential time course with a time constant of $2.8 \pm 0.8$ ms (with internal Cl⁻, at + 160 mV, $n = 10$ cells) to zero current levels (Fig. 2f). In symmetrical Cl⁻, VGLUT1$_{PM}$ fails to conduct Cl⁻ into the cell because it is closed at potentials positive to the reversal potential.

### VGLUT1 transports a variety of anions

To study VGLUT1$_{PM}$ currents carried by glutamate, we substituted Cl⁻ completely with glutamate in the pipette solutions of whole-cell experiments (Fig. 3a). To minimize intracellular contamination with external Cl⁻ ions, we only exposed cells to external Cl⁻ for limited recording times. Glutamate current amplitudes increased upon external acidification with a similar pH dependence to VGLUT1$_{PM}$ Cl⁻ currents (Fig. 3b) and also required external Cl⁻ (Supplementary Fig. 5a, b). As expected[22], VGLUT1$_{PM}$ glutamate currents are activated by external Cl⁻ and Br⁻, but not by I⁻ (Supplementary Fig. 5c).

Although VGLUTs selectively accumulate glutamate in synaptic vesicles[23,24], we recorded currents with similar characteristics in cells dialyzed with aspartate (Fig. 3c). We tested a variety of other polyatomic anions, such as gluconate, 2-(N-morpholino)ethanesulfonic acid (MES), methanesulfonic acid (MSA), isethionate, or HCO$_3^-$ (Fig. 3d–h); all of them generated similar currents as glutamate and aspartate (Fig. 3). For all tested polyatomic anions, VGLUT1$_{PM}$ currents are

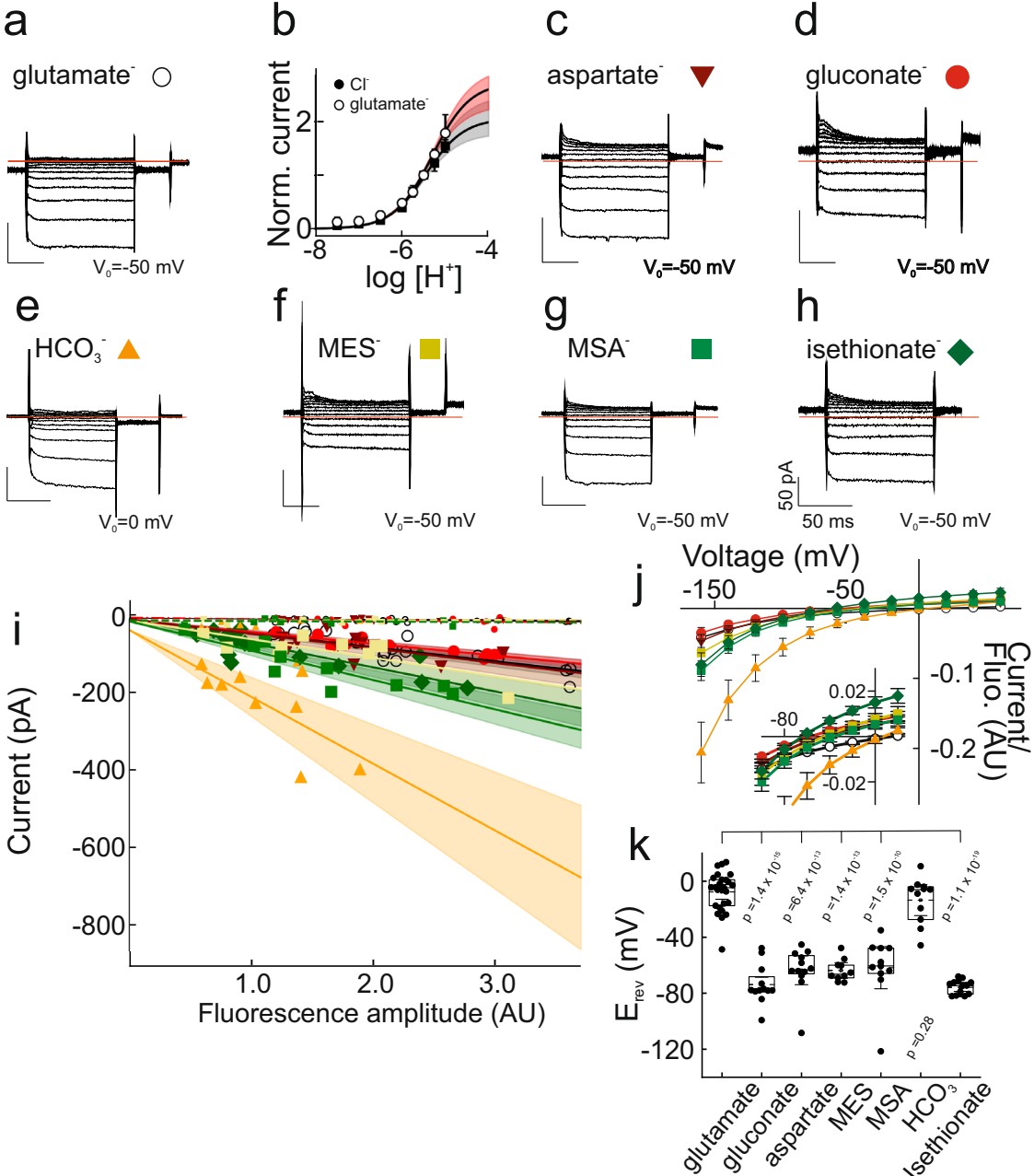

**Fig. 3 | VGLUT1$_{PM}$ mediates anion efflux/export of multiple large anions.**
**a**, **c**–**h** Representative measurements from cells dialyzed with different internal polyatomic anions at external pH 5.5 and 40 mM Cl⁻ after subtraction of background currents. Steps to voltages between −160 mV and +60 mV were applied from a holding potential of −50 mV. Red lines depict zero currents. **b** pH dependence of VGLUT1$_{PM}$ Cl⁻ and glutamate currents at −160 mV normalized to levels at pH 5.5. The solid red line indicates the bootstrapped mean of fits to Michaelis−Menten relationships, providing pK$_M$ = 5.2 ± 0.1 (means ± CI, $n$ = 10 cells) for glutamate. The black symbols/line provide the pH dependence of VGLUT1$_{PM}$ Cl⁻ currents from Fig. 1d. Shaded areas show the 95% CI. Cl⁻. **i** Whole-cell current amplitudes measured 100 ms after the voltage step to −160 mV increase linearly with whole-cell fluorescence. Each symbol provides fluorescence and current amplitudes from an individual cell; large symbols denote currents obtained at pH

5.5, small symbols at pH 7.4. Currents were not leakage substracted. Lines and shaded areas show the mean and 95% CI from linear fits. **j** Voltage dependence of current amplitudes measured 100 ms after the voltage step and normalized to the whole-cell fluorescence (means ± CI, $n$ = 26 (glutamate), 12 (gluconate), 12 (aspartate), 10 (MES), 10 (MSA), 11 (HCO$_3^-$), or 14 cells (isethionate)). The inset shows the same relationship for a limited voltage range, illustrating differences in current reversal potentials. **k** Current reversal potentials (E$_{rev}$) for multiple internal anions ($n$ = 26 (glutamate), 12 (gluconate), 12 (aspartate), 10 (MES), 10 (MSA), 11 (HCO$_3^-$), or 14 cells (isethionate). *P*-values from two-tailed *t*-test. Box plots show single data as symbols, mean values as dashed lines, medians as solid lines, upper and lower quartiles as box borders and 95% CI as whiskers. Source data are provided in the Source Data file.

absent at pH 7.4 and activated by external acidification. At pH 5.5 transfected cells exhibit much larger currents than control cells (Supplementary Fig. 2e, f). In control cells, background currents can be fully corrected by subtracting amplitudes at pH$_o$ = 7.4 from amplitudes at pH$_o$ = 5.5 (Supplementary Fig. 2g, h). For all tested polyatomic

anions, the comparison of multiple cells revealed a linear increase of whole-cell currents at −160 mV with whole-cell fluorescence under activating acidic conditions at pH 5.5, but not at pH 7.4 (Fig. 3i). We additionally tested block of glutamate, aspartate, gluconate, HCO$_3^-$, and MES currents by Rose Bengal, a noncompetitive VGLUT blocker

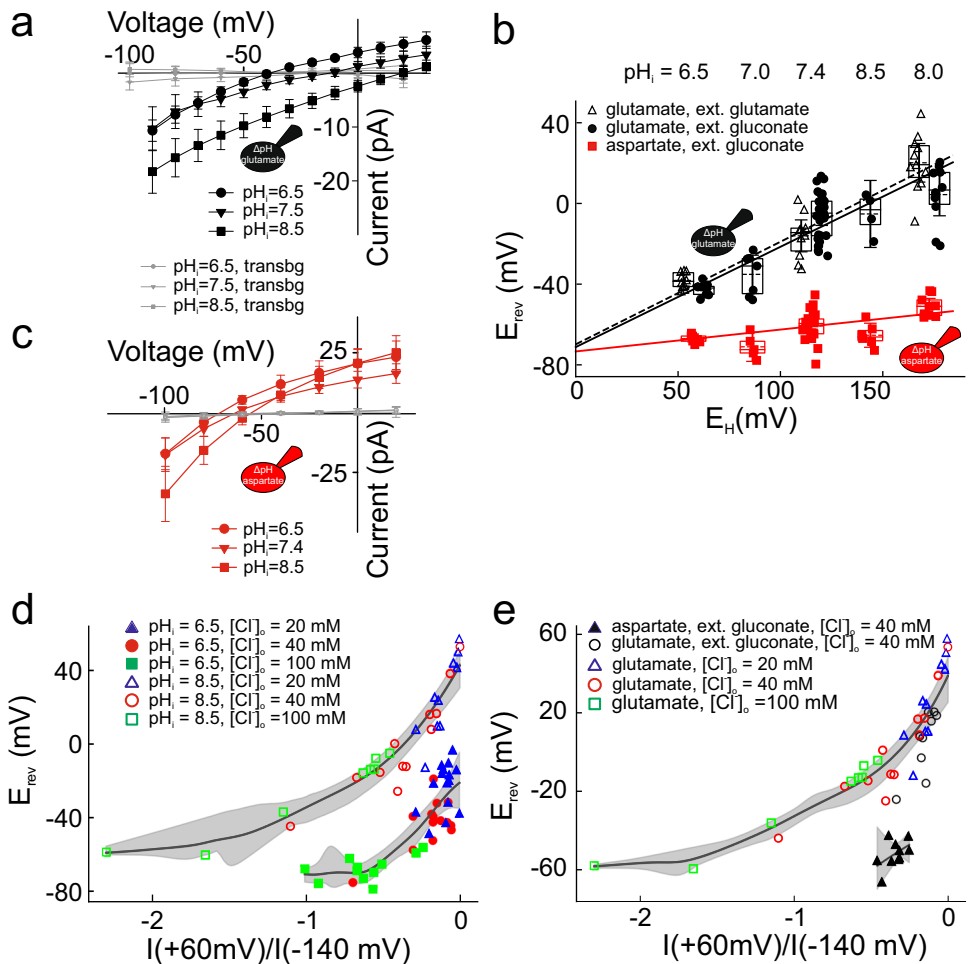

**Fig. 4 | VGLUT1_PM functions as H+-substrate exchanger with variable stoichiometry. a, c** Current–voltage relationships of glutamate (**a**) or aspartate (**c**) currents for various internal pHs after correction for background currents. The external solution contained 40 mM Cl⁻ and 100 mM glutamate (**a**)/gluconate (**c**) and was titrated to external pH 5.5. Small symbols show background currents (transbg) from cells expressing H191K-H426K-D428Q VGLUT1_PM under the same conditions. All data are given as means ± CI ($n = 10$ cells). **b** Changes of glutamate (filled circle, with an external solution containing 40 mM Cl⁻ and 100 mM gluconate ($n = 8, 7, 29, 4, 10$ cells), open triangle, with an external solution containing 40 mM Cl⁻ and 100 mM glutamate ($n = 10, 10, 11$ cells) and aspartate, filled square, with external solution containing 40 mM Cl⁻ and 100 mM gluconate ($n = 8, 5, 22, 6, 9$ cells) current reversal potentials with the Nernst potential for H⁺ calculated as $E_H = \frac{RT}{F} \ln \frac{[H^+]_o}{[H^+]_i}$. The external solution was titrated to external pH 5.5. Box plots show single data as symbols, mean values as dashed lines, medians as solid line, upper and lower quartiles as box borders, and 95% CI as whiskers. Solid and dashed lines represent weighted fitting to linear functions. **d** $E_{rev}$s as functions of relative Cl⁻ currents calculated as ratios of current amplitude at +60 mV by −140 mV. Closed symbols denote cells dialyzed with glutamate-based internal solution at $pH_i = 6.5$, open symbols at $pH_i = 8.5$. Solid lines represent the bootstrapped mean of non-parametric fits, and shaded area show bootstrapped 95% confidence intervals for those fits. Non-overlapping confidence intervals indicate a significant difference between $pH_i = 6.5$ and $pH_i = 8.5$ curves. **e** $E_{rev}$s as functions of relative Cl⁻ currents calculated as ratios of current amplitude at +60 mV by −140 mV. Closed black symbols denote cells dialyzed with aspartate-based internal solutions, open symbols with glutamate based solutions, all experiments were performed at $pH_i = 8.5$. At 40 mM $[Cl^-]_o$, aspartate reversal potentials were negative to glutamate reversal potentials ($p = 1.4 \times 10^{-8}$, two-tailed $t$-test), while relative Cl⁻ currents were not different. Solid lines represent the bootstrapped means of non-parametric fits, and shaded area show bootstrapped 95% confidence intervals for those fits. Non-overlapping confidence intervals indicate a significant difference between aspartate and glutamate curves. Source data are provided in the Source Data file.

with high affinity[25,26], and compared the results with block of VGLUT1_PM Cl⁻ or NO₃⁻ currents (Supplementary Fig. 2j, k). For all tested polyatomic anions, currents were blocked with comparable concentration dependence. We conclude that VGLUT1_PM can transport multiple anions.

Figure 3j depicts VGLUT1_PM current-voltage relationships for large anions after correction for leak currents and normalization to whole-cell fluorescence. In contrast to the absence of outward currents (corresponding to Cl⁻ influx) after dialysis with small anions, intracellular dialysis with large anions resulted in measurable positive currents that permitted quantification of the current reversal potential. This modification is due to alterations in the voltage dependence of VGLUT1_PM. VGLUT1_PM anion channels completely deactivate at

positive potentials in cells internally dialyzed with Cl⁻ (Fig. 2f), but not in cells with internal polyatomic anions (Fig. 3a, c–h). Supplementary Fig. 6 shows current responses to voltage protocols consisting of a hyperpolarizing voltage step to −140 mV followed by voltage steps to +120 mV of increasing durations and a fixed step to −120 mV. For Cl⁻-based internal solutions, membrane depolarization results in time-dependent channel deactivation that results in a decrease of current amplitudes measured directly after stepping back to −120 mV and subsequent current activation (Supplementary Fig. 6a, d). Addition of glutamate to the internal solution results in an almost complete absence of channel deactivation (Supplementary Fig. 6b, e). Control experiments reveal very small background currents also with 40 mM glutamate and 100 mM Cl⁻ in intracellular solutions (Supplementary

Fig. 6c). Cytoplasmic glutamate makes anion influx possible at positive potentials by impairing channel deactivation.

Current amplitudes observed for glutamate and aspartate, gluconate, MES, MSA and isethionate were comparable at negative voltages. However, whereas glutamate currents reversed at $-7.6 \pm 5.7$ mV ($n = 26$ cells), we observed less positive potentials for all other large anions (gluconate, $-73.8 \pm 8.9$ mV, $n = 12$ cells; aspartate, $-63.6 \pm 10.3$ mV, $n = 12$ cells; MES, $-63.7 \pm 5.6$ mV, $n = 10$ cells; MSA, $-61.8 \pm 15.0$ mV, $n = 11$ cells; isethionate, $-75.8 \pm 2.9$ mV, $n = 14$ cells; Fig. 3k); $HCO_3^-$ currents were much larger and reversed at $-13.5 \pm 11.0$ mV ($n = 11$ cells). We conclude that VGLUT1$_{PM}$ transports multiple large anions across the membrane, but with the highest driving force for glutamate.

## VGLUT1 mediates stoichiometrically coupled H⁺-glutamate exchange

Many vesicular neurotransmitter transporters function as H⁺-coupled exchangers[2]; however, H⁺-glutamate exchange by VGLUT remains under discussion[3,5,23,27–31]. Figure 4a depicts the voltage dependence of mean glutamate current amplitudes for different internal pHs at external glutamate-based solutions, with Cl⁻ present only in the external solution at 40 mM concentration. Increased internal pH reduces outward currents (caused by glutamate inward movement) and increases inward currents, resulting in a shift of the current reversal potential to more positive values. Control experiments illustrate negligible background currents under these ionic conditions for all tested pH$_i$. Figure 4b provides current reversal potentials ($E_{rev}$) as function of the Nernst potential for H⁺ ($E_H$) for VGLUT1$_{PM}$ glutamate currents for two different conditions, either with glutamate or with gluconate as main external anion. At pH$_i$ = 8.5 and glutamate on both membrane sides, reversal potentials ($20.2 \pm 9.5$ mV, $n = 11$) are positive to the predicted reversal potential for a glutamate uniporter (8.6 mV). The measured reversal potentials are neither compatible with glutamate uniport nor with choline-glutamate exchange by VGLUT1$_{PM}$. Current reversal potentials become more negative with increasing [Cl⁻]$_o$, opposite to the predictions of Cl⁻-glutamate exchange (Supplementary Fig. 7a). Taken together, these results indicate that VGLUT1$_{PM}$ functions as coupled H⁺-glutamate exchanger. In contrast, aspartate currents were only slightly affected by changes in pH$_i$ (Fig. 4c), resulting in only small changes in current reversal potentials under these conditions (Fig. 4b).

Coupled exchange of m glutamate for n H⁺ ions is at equilibrium at:

$$E_{rev} = \frac{RT}{(m+n)F}\left(mln\frac{[Glu^-]_i}{[Glu^-]_o} + nln(\frac{[H^+]_o}{[H^+]_i})\right) = \frac{m}{(m+n)}E_{Glu} + \frac{n}{(m+n)}E_H$$

$$(4)$$

with $E_{Glu}$ and $E_H$ being the Nernst potentials for glutamate ($E_{Glu}$) or H⁺ ($E_H$), F Faraday's constant, R the universal gas constant, and T the absolute temperature. Weighted linear regression to $E_H$-$E_{rev}$ plots (Fig. 4b) provides slopes of $0.51 \pm 0.1$ (internal and external glutamate) or $0.49 \pm 0.1$ (internal glutamate, external gluconate) and of $0.11 \pm 0.04$ for internal aspartate. The slope factors obtained for glutamate are consistent with a 1:1 coupling stoichiometry. The lower slope factor for aspartate suggests partial uncoupling of VGLUT1$_{PM}$ transport, i.e. approximately every eighth aspartate is transported in exchange for one proton, with all other aspartate ions transported in an uncoupled fashion. As expected for two transport processes that reach selectivity via distinct H⁺ coupling, extrapolation of the glutamate and aspartate current reversal potentials to $E_H = 0$ led to closely similar results (Fig. 4b).

Since VGLUTs require Cl⁻ to be functional[22], reversal potentials had to be measured in presence of external Cl⁻, resulting in VGLUT1$_{PM}$-mediated Cl⁻ influx and $E_{rev}$s negative to values given by Eq. 4. To

experimentally assess how Cl⁻ currents affect the coupling stoichiometries obtained from $E_{rev}$-$E_H$ plots, we measured glutamate currents for various pH$_i$ and [Cl⁻]$_o$ and plotted $E_{rev}$s versus relative Cl⁻ current amplitudes (determined as ratios of current amplitudes at +60 mV by values at −140 mV) (Fig. 4d). Changes in pH$_i$ neither modify absolute (Supplementary Fig. 7b) nor relative anion current amplitudes (Fig. 4d). At fixed pH$_i$ current reversal potentials shift upon increasing Cl⁻ current amplitudes in a monotonic relation. However, $E_{rev}$-Cl⁻ currents relationships are clearly separate for pH$_i$ 6.5 or pH$_i$ 8.5. At the same Cl⁻ current amplitudes, currents at pH$_i$ 6.5 reverse at more negative potentials than for pH$_i$ 8.5 (Fig. 4d). These results demonstrate that pH$_i$-dependent changes of $E_{rev}$ are caused by altered driving forces for glutamate transport and not by changes in Cl⁻ channel activity. The corresponding $E_{rev}$-Cl⁻ current plot for internal aspartate provides—at comparable relative Cl⁻ currents—more negative reversal potentials than for glutamate currents (Fig. 4e), indicating that the reduced slope factor for VGLUT1$_{PM}$ aspartate transport is not due to higher Cl⁻ channel activity, but to partial uncoupling.

We additionally fitted current-voltage relationships at various pH$_i$ and [Cl⁻]$_o$ with sums of H⁺-glutamate exchange currents (at 1:1 stoichiometry) and Cl⁻ currents predicted by the Goldman-Hodgkin-Katz equation (Supplementary Fig. 7c). $E_{rev}$s obtained from such fits are in good agreement with experimental data (Supplementary Fig. 7d). Increasing [Cl⁻]$_o$ decreases the slopes of linear fits to $E_H$-$E_{rev}$ plots. We conclude that—as every leak conductance in such analyses—VGLUT1$_{PM}$ Cl⁻ currents may cause underestimation of coupling coefficients obtained from these slopes. Since we used reversal potentials at [Cl⁻]$_o$ = 40 mM (Fig. 4b) to calculate coupling stoichiometries, VGLUT1$_{PM}$ might exchange H⁺ and glutamate at ratios even higher than 1:1.

Current-voltage relationships from cells with glutamate as main internal show only slightly larger current amplitudes with external gluconate than with external glutamate (Supplementary Fig. 7e), at only marginally different $E_{rev}$s (Fig. 4b). For these two conditions, I-V plots can be well fit with sums of currents predicted for a coupled glutamate-H⁺ exchanger and passive Cl⁻ and gluconate influx, either without [gluconate]$_o$ or with negligible [glutamate]$_o$ (solid lines, Supplementary Fig. 7e). Substitution of external glutamate with gluconate results in a largely increased driving force for glutamate outward transport. This change does not result in significant shifts of the reversal potential (Fig. 4b), since the currents in the other direction are also increased: gluconate inward transport is passive, whereas glutamate inward transport occurs against a 100-fold H⁺-gradient.

We next determined unitary glutamate transport rates from macroscopic current–fluorescence plots for Cl⁻ and glutamate currents (Supplementary Fig. 5d). Since transporters are expressed as fluorescent fusion proteins, the number of transporters can be quantified in each cell from its whole-cell fluorescence. Only a percentage of the transporters localizes to the surface membrane, however, unitary current amplitudes and absolute open probabilities obtained by noise analysis for VGLUT1$_{PM}$ Cl⁻ channel activity (Fig. 2c, d) can be used to convert whole-cell fluorescences into numbers of transporters in the surface membrane. We multiplied the ratio of slopes of macroscopic current–fluorescence plots for glutamate and Cl⁻ currents with averaged unitary Cl⁻ currents, i.e. the product of unitary current amplitudes and absolute open probabilities obtained by noise analysis (5.8 fA), and obtained unitary glutamate current amplitudes of 0.18 fA. Division of this unitary glutamate current amplitude by the two elementary charges that are translocated in one H⁺-glutamate exchange cycle results in a transport rate of around $561 \pm 123$ s⁻¹ at −160 mV.

## Coupled H⁺-glutamate exchange is necessary for effective glutamate accumulation in synaptic vesicles

To understand how differences in H⁺ coupling affect the vesicular neurotransmitter accumulation, we developed a continuum

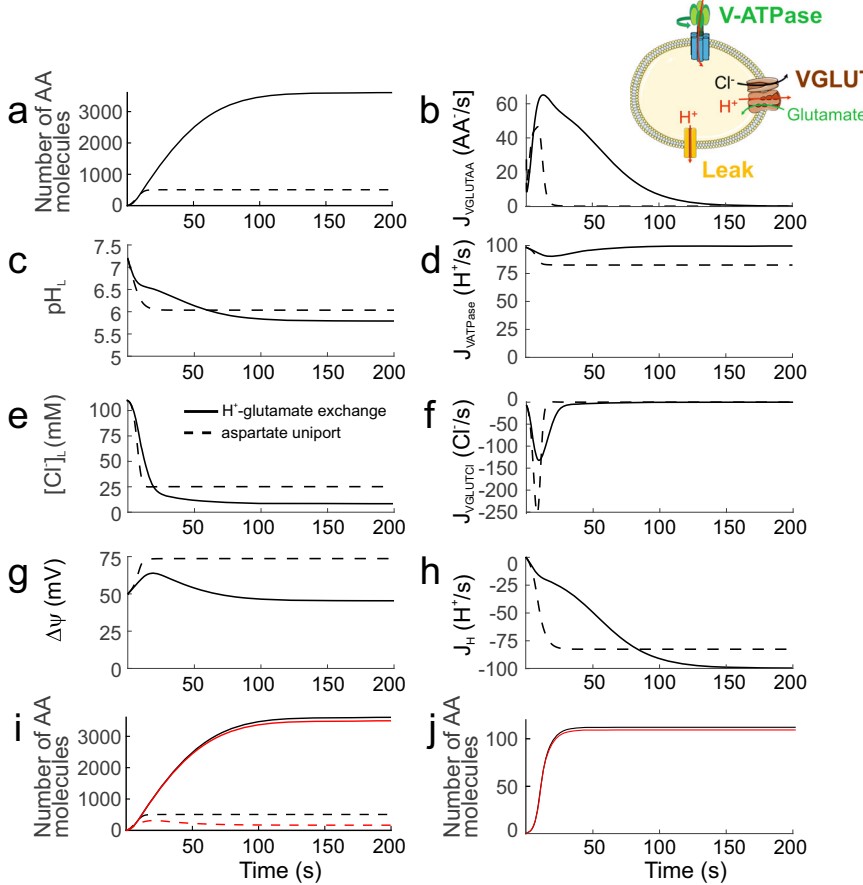

**Fig. 5 | A continuum quantitative model of synaptic vesicles that describes acidification, substrate accumulation, and Cl⁻ efflux for H⁺-coupled and uncoupled transport. a–h** Predicted evolution of luminal amino acid (AA) numbers (**a**), pH (**c**), [Cl⁻] (**e**), luminal membrane potential (**g**), glutamate/aspartate transport rates (**b**), active proton transport rates (**d**), Cl⁻ currents by VGLUT chloride channels (**f**), and proton leak fluxes (**h**) for vesicles taking up glutamate via H⁺-glutamate exchange (solid lines) or aspartate via an aspartate uniporter (dashed lines). **i** Comparison of luminal glutamate (solid lines) and aspartate (dashed lines) numbers in the presence of both amino acids at the external side (red) or with only glutamate or aspartate present (black). [glutamate] and [aspartate] were set either to 0 or 10 mM. **j** Glutamate accumulation with external 50 μM glutamate alone (black line) or together with 1 mM aspartate (red line). Transport molecules within the model vesicle are schematically shown as inset.

quantitative model of synaptic vesicles that describes acidification, substrate accumulation, and Cl⁻ efflux using a set of differential equations[32,33]. We studied two VGLUT transport modes: stoichiometrically coupled H⁺-glutamate exchange and aspartate uniport as oversimplification of the partial uncoupling by aspartate (Fig. 5). The model vesicle was acidified by V-type ATPases and contained H⁺ leak channels to stabilize the vesicular pH during sustained proton pumping[3] (Fig. 5). Glutamate/aspartate and Cl⁻ transport by the VGLUTs were described with separate sets of differential equations for H⁺-glutamate exchange/aspartate transport and for chloride channel function. Modelled exchanger and channel transport displayed identical pH and Cl⁻ dependences; for simplicity, transport rates/Cl⁻ currents were assumed to change linearly with the driving force. As starting conditions, we chose high [Cl⁻], neutral pH and negligible glutamate/aspartate concentrations in the synaptic vesicle, resembling the initial conditions after endocytosis.

Figure 5 shows predicted evolution of luminal glutamate (solid lines) or aspartate (dashed lines) (Fig. 5a), pH (Fig. 5c), [Cl⁻] (Fig. 5e), the luminal membrane potential ΔΨ (Fig. 5g), glutamate/aspartate transport rates (Fig. 5b), active proton transport rates by V-type ATPases (Fig. 5d), Cl⁻ currents mediated by VGLUT chloride channels (Fig. 5f), and proton leak fluxes (Fig. 5h). H⁺-glutamate exchange mediates much higher transport rates than aspartate uniport (Fig. 5a), predicting highly selective glutamate uptake by synaptic

vesicles. Moreover, Cl⁻ removal from the vesicle is more effectively supported by H⁺-glutamate exchange than by aspartate uniport (Fig. 5e, f). In contrast, we observed only small differences in associated V-type ATPase-mediated H⁺ accumulation rates (Fig. 5d) or H⁺ leak currents (Fig. 5h), when comparing H⁺-glutamate exchange and aspartate uniport. H⁺-glutamate exchange depolarizes the vesicular membrane potentials less than amino acid uniport (Fig. 5g) and thus reduces the electrochemical gradient for H⁺ accumulation. H⁺-glutamate exchange was sustained after reaching steady-state pH (Fig. 5b), in agreement with experimental results[34]. The model also correctly predicts selective glutamate accumulation under physiological conditions in the presynaptic nerve terminal, i.e. with equal cytoplasmic amounts of glutamate and aspartate (Fig. 5i). In radiotracer flux measurements, an excess of external aspartate fails to reduce radioactive glutamate uptake[23]. The model also correctly reproduces experimental results under such ionic conditions (Fig. 5j).

Glutamate accumulates in model vesicle with a time constant of 36 s, which is in the range of experimental results[35,36]. Supplementary Fig. 8 provides changes in predicted filling time constants upon variation of vesicular glutamate transporter numbers for different numbers of V-type ATPases per vesicle. Increased numbers of VGLUTs sharply decrease time constants to minimum values with three to five transporters per vesicle. Filling time constants of vesicles with a single

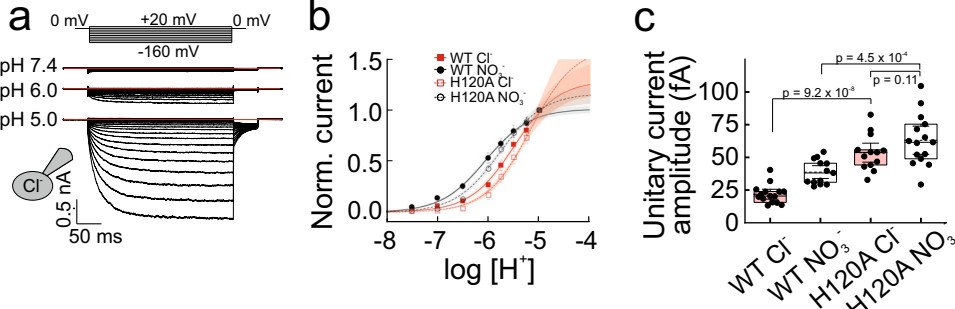

**Fig. 6 | H120A changes the anion channel properties of VGLUT1$_{PM}$.**
**a** Representative whole-cell recordings from HEK293T cells transiently transfected with H120A VGLUT1$_{PM}$. Cl$^-$ was the internal permeable anion, and cells were perfused externally with Cl$^-$-based solutions of different pH values. **b** pH dependence of WT (filled symbols) and H120A (open symbols) VGLUT1$_{PM}$ anion currents at −160 mV. Solid red line indicates the bootstrapped mean of fits to Michaelis–Menten relationships, providing pK$_M$ = 5.2 ± 0.08 with Cl$^-$ (means ± CI,

$n = 17$ cells, red); pK$_M$ = 5.8 ± 0.04 with NO$_3^-$ ($n = 15$ cells, black)); two-tailed $t$-test, $p = 0.00181$. Shaded areas show the 95% CI. The black symbols/line provide the pH dependence of VGLUT1PM Cl- currents from Fig. 1d. **c** Comparison of WT and H120A VGLUT1$_{PM}$ single channel amplitudes for Cl$^-$ ($n = 13$) and NO$_3^-$ ($n = 15$). $P$-values from two-tailed $t$-test. Box plots show single data as symbols, mean values as dashed lines, medians as solid lines, upper and lower quartiles as box borders and 95% CI as whiskers. Source data are provided in the Source Data file.

VGLUT decrease from 36 s to 12.7 s, when the average number of V-ATPase complexes per vesicle is increased from 1 to 3.

We next used the model to assess the effect of the VGLUT chloride channel function on vesicular glutamate loading. Supplementary Fig. 9 presents time courses of luminal [glutamate], pH, [Cl$^-$], and membrane potential and of glutamate transport rates and active proton transport rates for the synaptic vesicle model established in Fig. 5 at increased or decreased relative VGLUT Cl$^-$ currents. The complete blockade of VGLUT channel-mediated Cl$^-$ currents resulted in less-pronounced depolarization, thereby reducing glutamate accumulation at increased ATP consumption for primary active H$^+$ transport. Reduced VGLUT Cl$^-$ currents led to lower glutamate transport rates, again at increased levels of primary active H$^+$ transport. These results demonstrate a functional role for VGLUT Cl$^-$ channels beyond supporting osmotically neutral glutamate accumulation[29]. We conclude that VGLUT anion-channel-mediated Cl$^-$ efflux supports the energy efficiency of vesicular glutamate accumulation.

### H120A substitution uncouples H$^+$-glutamate exchange
Substitution of histidine at position 128 by alanine was reported to drastically reduce VGLUT2-mediated glutamate transport[30]. The corresponding H120A mutation into VGLUT1$_{PM}$ (Supplementary Fig. 1c) leaves many features of VGLUT1$_{PM}$ anion currents unaffected (Fig. 6a): currents exhibit strong rectification, with absent currents at positive potentials and hyperpolarization-induced activation. H120A VGLUT1$_{PM}$ channels are closed at neutral pH and open with a pH dependence, which was shifted toward more acidic values (Fig. 6b). H120A and WT VGLUT1$_{PM}$ differed in their time course of activation, with slower activation for H120A VGLUT1$_{PM}$.

Noise analysis revealed larger single channel amplitudes for H120A VGLUT1$_{PM}$ anion channels (52 ± 3.0 fA ($n = 13$ cells)) with internal Cl$^-$ than for WT VGLUT1$_{PM}$ under the same conditions (25 ± 2 fA ($n = 15$ cells, two-tailed $t$-test, $p = 9.21 \cdot 10^{-8}$); Fig. 6c). H120A VGLUT1$_{PM}$ single channel amplitudes with internal NO$_3^-$ were not increased compared with internal Cl$^-$ (63 ± 2 fA ($n = 15$ cells, two-tailed $t$-test, $p = 0.11$)). Figure 7a, b show representative current recordings of H120A VGLUT1$_{PM}$ anion currents for various small and large anions. H120A changes the relative I$^-$ conductance and, more importantly, shifts the VGLUT1 current reversal potentials measured in cells dialyzed with glutamate$^-$ or aspartate$^-$ toward more negative potentials (Fig. 7c−e). No changes in the reversal potentials were observed in experiments with internal gluconate$^-$ or HCO$_3^-$ (Fig. 7e). In experiments resembling those used to study the transport stoichiometry of H$^+$-glutamate exchange by WT VGLUT1$_{PM}$ (Fig. 4), we did not observe

any shifts in reversal potential with changing intracellular pH, indicating that H120A abolishes H$^+$ coupling in VGLUT1 (Fig. 7f).

## Discussion
We found that VGLUT1 functions as a coupled transporter that exchanges H$^+$ and glutamate at 1:1 ratio or higher (Fig. 4). In experiments with aspartate, reversal potentials were less affected by varying transmembrane ΔpH, indicating that VGLUT1 aspartate transport is partially uncoupled from the transmembrane proton gradient (Fig. 4). VGLUT1 transport of Cl$^-$ and NO$_3^-$ is not coupled to H$^+$ transport across the membrane[6], and we demonstrate that VGLUT1 functions as Cl$^-$ and NO$_3^-$ channel (Fig. 2). Mathematical modeling demonstrated that the distinct transport mechanisms are sufficient for selective glutamate accumulation at physiological ion and substrate concentrations (Fig. 5). H120A impairs VGLUT glutamate uptake[30] by uncoupling glutamate and H$^+$ transport (Fig. 7). These results define VGLUT1 as H$^+$-coupled glutamate exchanger that attains its selectivity for different substrates via varying the transport stoichiometry.

VGLUTs are currently assumed to function as glutamate uniporters[1,3,4], based on two lines of evidence. In reconstituted systems, glutamate uptake can be driven by electric gradients without pH gradients, but not by pH gradients alone[30,37]. However, this result can be explained by the strong voltage dependence of VGLUT glutamate transport (Fig. 3[6]): low transport rates at voltages around 0 mV make it difficult to detect changes in activity upon varying ΔpH. Additional experimental support for exclusively voltage-driven VGLUT transport was obtained from imaging experiments that demonstrated lower proton efflux rates for glutamatergic than for GABAergic synaptic vesicles[9,36]. Since absolute transport rates of VGATs have not yet been determined and little is known about vesicular H$^+$ or HCO$_3^-$ leak current, we believe that these experiments leave space for coupled exchange combined with additional anion channel activity by VGLUTs.

Our experiments were performed with mutant transporters carrying substitutions in cytoplasmic domains. VGLUT1$_{PM}$ transport functions fully account for earlier work with vesicle preparations, and VGLUT1$_{PM}$ whole-cell currents closely resemble whole-cell or whole oocyte currents of another surface membrane-optimized mutant VGLUT[6] and whole endosome currents of WT VGLUT1[7]. All these results indicate that the amino acid substitutions in VGLUT1$_{PM}$ do not cause major changes in transport functions. To test for coupled H$^+$-glutamate exchange, we measured the pH$_i$ dependence of current reversal potentials (Fig. 4)[38,39]. We performed experiments in an expression system with high expression levels, and control of extra- and intracellular ions. In all experiments, currents were corrected for

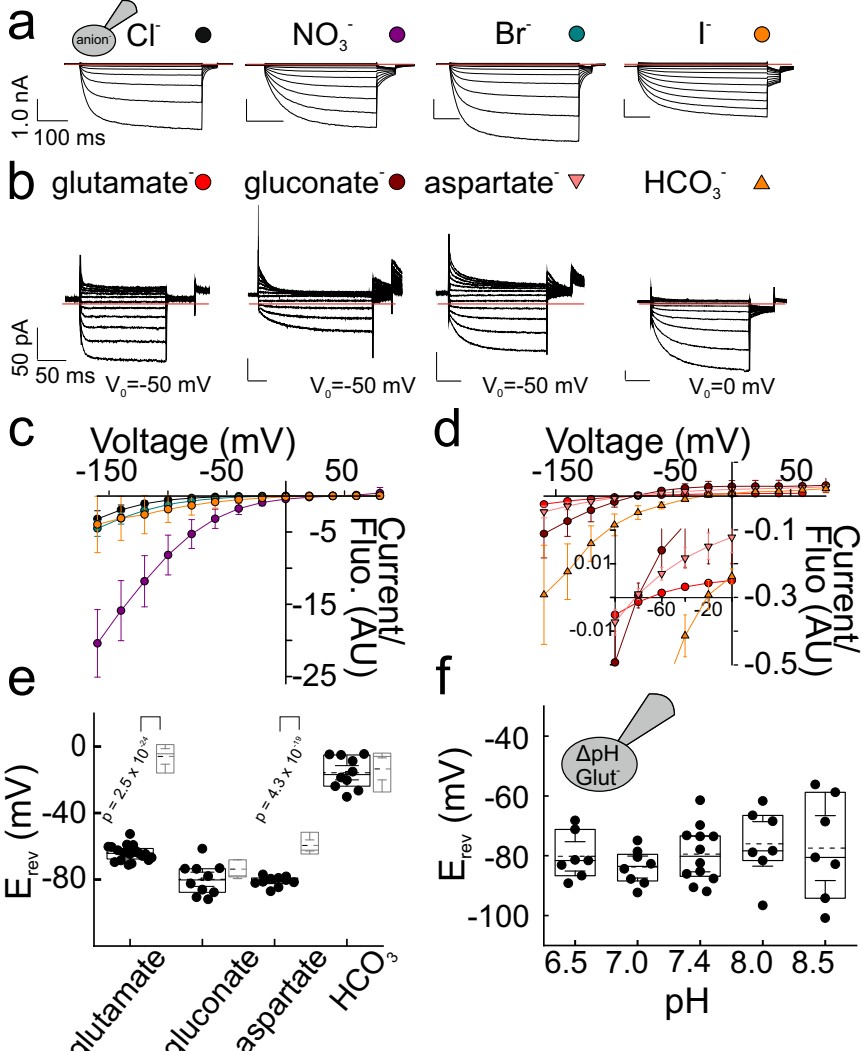

**Fig. 7 | H120A abolishes H⁺-glutamate coupling of VGLUT1$_{PM}$. a, b** Representative whole-cell recordings of HEK293T cells transiently transfected with H120A VGLUT1$_{PM}$. Cells were internally dialyzed with the indicated small (**a**) or large (**b**) anions, held to 0 mV (**a**) or −50 mV (**b**). After stepping to various voltages, a fixed step to − 50 mV was applied at the end of each recording. Background currents determined in the same cell at alkaline pH were subtracted. Red lines depict zero currents. **c, d** Voltage dependence of current amplitudes measured 100 ms after the voltage step and normalized to the whole-cell fluorescence for small (means ± CI, $n$ = 11 (Cl⁻), 11 (NO₃⁻) 10 (I⁻), 10 (Br⁻)) or large polyatomic ($n$ = 20 (glutamate), 10 (gluconate), 10 (aspartate), 10 cells (HCO₃⁻)) anions. The inset shows a magnified image of the currents in the region of their reversal potentials, presented individually in (**e**). **e** Current reversal potentials ($E_{rev}$) obtained for H120A (black, $n$ = 20, 10, 10, 10) or WT (grey) for several internal anions. Box plots show single data as symbols, mean values as dashed lines, medians as solid lines, upper and lower quartiles as box borders and 95% CI as whiskers. **f** Dependence of current reversal potentials for H120A VGLUT1$_{PM}$ glutamate currents on the internal pH ($n$ = 7, 8, 13, 7, 7 cells. $P$-values from two-tailed $t$-test. Box plots show single data as symbols, mean values as dashed lines, medians as solid lines, upper and lower quartiles as box borders and 95% CI as whiskers. Source data are provided in the Source Data file.

background current components, and VGLUT1$_{PM}$–like currents were neither observed in non-transfected cells nor in cells expressing a non-functional VGLUT1$_{PM}$ mutant (Supplementary Fig. 2). H120A did not affect current amplitudes, but modified the time and voltage dependence of VGLUT1$_{PM}$ currents for all internal anions. Moreover, H120A specifically changed the reversal potential and the pH$_{int}$ dependence of glutamate currents, but not of any other current component. Thus, we are confident that the observed currents are transported by VGLUT1$_{PM}$ and not by endogenous proteins.

VGLUTs are not only glutamate transporters, but also Cl⁻ channels, and this dual function prevents measurements of reversal potentials of glutamate transport currents in isolation. However, at certain pH$_i$, we observed current reversal potentials positive to values predicted for glutamate uniporters also in the presence of VGLUT1$_{PM}$-mediated Cl⁻

influx (Fig. 4b). Glutamate current reversal potentials measured for various pH$_i$ change linearly with $E_H$ with a slope expected for a 1:1 exchange stoichiometry (Fig. 4b). These shifts of glutamate reversal potentials with pH$_i$ and differences in current reversal potentials between glutamate and aspartate are not caused by changes in Cl⁻ channel activity (Fig. 4d, e). As every leakage current in transport stoichiometry assessment via reversal potential measurements, VGLUT1$_{PM}$ Cl⁻ currents reduce slopes of $E_H$-$E_{rev}$ relationships (Supplementary Fig. 7d), so that transport stoichiometries might be underestimated. Taken together, these results indicate that WT VGLUT1$_{PM}$ mediates coupled H⁺-glutamate exchange at 1:1 stoichiometry (or even higher) and that aspartate and H⁺ transport are partially uncoupled. H120A reduces glutamate uptake[30] via abolished proton coupling (Fig. 7), leaving all other hallmark features of VGLUT1 unaltered.

We observed similar VGLUT1$_{PM}$ current amplitudes for every tested polyatomic anion: not only for glutamate, but also for aspartate, gluconate, MES, MSA and isethionate (Fig. 4). However, only glutamate transport is coupled to H$^+$, resulting in much higher driving forces for glutamate than for other anions. The distinct transport coupling suffices for selective glutamate accumulation[40–42] (Fig. 5). Moreover, H120A abolishes glutamate uptake in radiotracer experiments[30] by uncoupling H$^+$ and glutamate transport (Fig. 7). We did not study whether the VGLUTs additionally distinguish between anions that are transported in the uniporter mode. Although there must exist anions that are too large to translocate via VGLUTs, size limits are difficult to predict. The structurally related SLCO and SLC22A transporters are capable to transport a variety of large anions[43]. We therefore decided to restrict the analysis to the tested six anions; we feel that this analysis convincingly demonstrates how VGLUTs utilize variable substrate stoichiometry to attain glutamate selectivity.

Mathematical modelling demonstrated how the three main features of VGLUT, coupled H$^+$-glutamate transport, Cl$^-$ channel function and pH activation, optimize synaptic vesicle filling. It does not only optimizes neurotransmitter accumulation (Fig. 5a), but also reduces the membrane potential of the synaptic vesicle (Fig. 5g) and thus improves the efficiency of the V-type ATPase[44] (Fig. 5c). Cl$^-$ channel function accelerates glutamate accumulation (Supplementary Fig. 9), and Cl$^-$ channel function together with less positive synaptic membrane potential due to coupled H$^+$-glutamate transport minimizes resting [Cl$^-$]$_{SV}$ (Fig. 5e), permitting osmotically neutral glutamate accumulation. Coupled H$^+$-glutamate exchange increases the driving force for vesicular glutamate uptake by harnessing vesicular H$^+$ concentration gradients and by exploiting the vesicular potential more effectively compared with uniporters, with a twofold increase in the number of moved charges (Fig. 5).

Synaptic vesicles exhibit high internal [Cl$^-$] after endocytosis; thus, effective vesicular glutamate accumulation requires the substitution of Cl$^-$ by glutamate[29]. However, VGLUT1$_{PM}$ currents exhibit a pronounced inward rectification (Fig. 1) that will prevent Cl$^-$ efflux out of the vesicle under many conditions (Fig. 1). Rectification is due to voltage-dependent gating: steps toward positive voltages elicit positive currents after activation by membrane hyperpolarization that deactivate over a very short time course (Fig. 2f). Cytoplasmic anions modify VGLUT1 gating, and cytoplasmic glutamate (as for other large polyatomic anions) keeps the channel open (Fig. 3, Supplementary Fig. 6), permitting Cl$^-$ flux in the opposite direction (Fig. 3). H120A abolishes H$^+$-glutamate coupling while supporting glutamate currents (Fig. 6). Moreover, H120A VGLUT1$_{PM}$ robustly conducts anions currents with increased unitary current amplitudes (Fig. 6), with gating still tightly coupled to the external pH. Since H120A abolishes H$^+$-glutamate exchange, but only slightly affects proton activation, proton coupling and proton activation are likely to involve distinct protonation processes.

In summary, we demonstrate that VGLUTs are H$^+$-anion exchangers that exploit a variable transport stoichiometry for transport selectivity: only glutamate transport is tightly coupled to H$^+$ transport, making VGLUTs selective for glutamate over other cytoplasmic anions. Cl$^-$ permeates through the VGLUTs independently of H$^+$ transport. The resulting differences in driving force permit Cl$^-$ efflux during glutamate accumulation and, thus, osmotically neutral glutamate accumulation. VGLUT glutamate transport and Cl$^-$ conduction require acidic luminal pH, and both are activated by luminal positive membrane potentials. Such regulation by both pH and voltage ensures selective glutamate import by permitting VGLUT transport/conduction only under conditions with a much higher driving force for glutamate than for other anions. Thus, variable H$^+$-coupling permits selective and energy-efficient VGLUT-mediated glutamate accumulation in synaptic vesicles.

## Methods

### Cell lines
HEK293T cells (Sigma-Aldrich) were transiently transfected using a lipofectamine precipitation method and examined 24 h later. Stably transfected Flp-In T-Rex 293 were used after induction with 1.5 µg/ml tetracycline for a maximum of 24 h[45]. HEK293T and Flp-In T-Rex 293 cultures were maintained in humidified incubators at 37 °C and 95% air/5% CO$_2$ in culture medium (Dulbecco's modified Eagle Medium containing 10% FBS and supplemented with penicillin and streptavidin).

### Expression plasmids, mutagenesis, and heterologous expression
To express VGLUT1 as an eGFP fusion protein, we inserted the coding region of eGFP into pcDNA3.1-rVGLUT1 (kindly provided by Dr. Shigeo Takamori) at the 3′ end of the transporter coding region. Using a method similar to Eriksen et al.[6], we mutated various dileucine-like endocytosis motifs to alanine to promote plasma membrane insertion (Supplementary Fig. 1) using PCR-based techniques. Transiently expressed EE6,7AA/L11A/EE505,506AA/F510A/V511A VGLUT1-GFP (VGLUT1$_{PM}$) predominantly localized to the surface membrane (Supplementary Fig. 1). H120A or the triple mutant H191K-H426K-D428Q were introduced into VGLUT1$_{PM}$ using PCR-based techniques. To generate inducible cell lines, we subcloned rVGLUT1-eGFP into pcDNA5/FRT/TO.

### Whole-cell patch clamp and fluorescence measurements
Standard whole-cell patch clamp recordings were performed using EPC 10 amplifiers, controlled by PatchMaster (HEKA Elektronik, Germany)[46]. We used borosilicate pipettes (Harvard Apparatus, USA) with resistances of 0.9–3 MΩ and applied series resistance compensation and capacitance cancellation, resulting in a voltage error of <5 mV. The standard bath solution contained (in mM) 136 choline chloride, 2 MgCl$_2$, and 30 HEPES, and we adjusted the osmolarity of the bath solution with glucose to a value of 15 mOsm/l higher than the internal solution. For external and internal pH values between 6.5 and 8.5, HEPES was used as the buffer and adjusted with choline hydroxide; for more acidic perfusion solutions, 30 mM HEPES was replaced with 50 mM MES and the pH was adjusted with choline hydroxide. Experiments with an internal pH of <6.5 activated endogenous currents, which prevented the reliable determination of VGLUT1$_{PM}$ currents under these conditions. In experiments with large polyatomic anions in the pipette solution, 100 mM choline chloride was replaced by choline gluconate in the bath solution; and cells were held at −50 mV. In some of the experiments to assess the transport stoichiometry, gluconate was fully substituted by glutamate (Fig. 4a), and in experiments with variable [Cl$^-$]$_o$ (Fig. 4; Supplementary Fig. 6), [Cl$^-$]$_o$ was adjusted by exchanging Cl$^-$ with glutamate$^-$ or vice versa. The standard pipette solution contained (in mM) 140 choline anion$_X$, 5 EGTA, 5 Mg(OH)$_2$, and 30 HEPES, adjusted to pH 7.4 with TMA-OH (with anion$_X$ = Cl$^-$, NO$_3^-$, Br$^-$, I$^-$, HCO$_3^-$, glutamate$^-$, aspartate$^-$, gluconate$^-$, MES$^-$, MSA$^-$ or isethionate$^-$). Solutions containing HCO$_3^-$ were oxygenated with carbogen (5% CO$_2$ in O$_2$) until the start of the experiment. We used external and/or internal agar salt bridges (made from plastic tubing filled with 0.5 M KCl in 2% agar) to connect the Ag/AgCl electrode. Junction potentials were used to correct the results. VGLUT1$_{PM}$ currents are much larger than background for all tested anions, and VGLUT1$_{PM}$ currents are blocked by Rose Bengal (Supplementary Fig. 3). Unless reported otherwise, current recordings were corrected for leakage currents by subtracting current amplitudes obtained in the same cell with external pH 7.4 or pH 8.5 under identical conditions. To support complete intracellular dialysis, we waited at least 2 min after establishing the whole-cell mode before starting electrical recordings.

In experiments combining fluorescence measurements and whole-cell recordings, eGFP was excited using a Polychrom V

monochromator (TILL Photonics) set to 488 nm and recorded using a Neo 5.5 sCMOS (Andor)[47]. Mean fluorescence intensities were determined for the whole cell body using Fiji (http://www.fiji.sc) and given as arbitrary units (AU). Since transporters are expressed as fluorescent fusion proteins, whole-cell eGFP fluorescences are proportional to the number of transporters per cell. We did not distinguish fluorescences of intracellular compartments and the surface membrane. Cells are always imaged before establishing the whole-cell mode, and we assume that every tested cell exhibits a similar subcellular distribution of transporters and that the whole-cell fluorescence provides a value proportional to the number of transporters in the surface membrane[11,12,47,48]. Possible limitations of this approach are changes in the percentage of transporters in the surface membrane because of saturation of the intracellular trafficking machinery[49] and inaccuracies due to predominant intracellular localizations of the protein under study[50]. For VGLUT1$_{PM}$, there are no indications for saturation of whole-cell currents with increasing expression (Figs. 1b and 3i), and VGLUT1$_{PM}$ mostly resides in the surface membrane (Supplementary Fig. 1).

### Noise analysis

For noise analysis, currents were digitized with a sampling frequency of 100 kHz, and power spectra were calculated from steady-state currents during 50-s voltage jumps to −100 mV (46 s analyzed) using fast Fourier transformation. Data were binned into 2300 data points, averaged over all measured cells, and fitted by the sum of double Lorentzian function and a term representing pink noise:

$$S = \frac{S_1}{1 + \frac{f}{fc_1}} + \frac{S_2}{1 + \frac{f}{fc_2}} + \frac{S_3}{f^a} \tag{5}$$

with S denoting the spectral density, f the frequency, a the slope of the linear pink noise, and $S_x$ and $f_{cx}$ the amplitude and corner frequencies.

In principle, proton activation of VGLUT1$_{PM}$ can be caused by pH-dependent unitary current amplitudes or pH-dependent open probabilities. The current variance after correction for background variance (measured at neutral/alkaline pH) ($\sigma^2 - \sigma_{bg}^2$) depends on unitary current amplitudes i and absolute open probabilities.

$$\frac{\sigma^2 - \sigma_{bg}^2}{I} = i(1 - p) \tag{6}$$

Supplementary Fig. 4 depicts a plot of $\frac{\sigma^2 - \sigma_{bg}^2}{I}$ against the external pH, fitted with Eq. 2 either under the assumption either that only the open probability changes with pH and the single channel amplitude remains constant (red line)

$$\sigma^2(pH) - \sigma_{bg}^2 = i^2 N p_{pH}(1 - p_{pH}) \tag{7}$$

or that unitary current amplitudes change with pH and the open probability remains constant (blue line).

$$\sigma^2(pH) - \sigma_{bg}^2 = i_{pH}^2 N p(1 - p) \tag{8}$$

We obtained the pH dependence of p or i from the pH dependence of the macroscopic current amplitude for each of the two assumptions and fitted the pH dependence of the variance with either Eq. 7 or Eq. 8. We then computed $\frac{\sigma^2 - \sigma_{bg}^2}{I}$ using the obtained fit parameter and plotted the experimental and fitted values against pH (Supplementary Fig. 4).

Unitary current amplitudes were determined by analyzing current noise over a period of 15 ms 100 ms after the voltage step to −160 mV (indicated in red in Fig. 2b) at five to eight different external pH values between 5.0 and 7.4. Since steady-state conditions were not always

reached during these recording times, high frequency noise signals were separated from slowly relaxing current amplitude traces by applying a Butterworth high-pass filter with a cut-off frequency of 200 Hz. The obtained σ²−I plots did not span a full parabola, so we applied F-test model selection on all data sets obtained under each condition (internal Cl⁻ or NO₃⁻) to assess whether they could best be described by a parabola or a linear function. For both conditions, a parabolic function provided a significantly better fit than a linear function. Consequently, we calculated single channel amplitudes from parabolic fits:

$$\sigma^2 = iI - \frac{I^2}{N} + \sigma_{bg}^2 \tag{9}$$

To estimate the duration of the single-channel events, we simulated single-channel events from a two-state model with different open times, and compared the normalized standard deviations for currents from excised outside-out recordings filtered at different frequencies with the standard deviations for simulated currents[15,21].

### Modeling glutamate accumulation in synaptic vesicles

We developed a continuum quantitative model of synaptic vesicles that describes the temporal evolution of luminal glutamate/aspartate molecule numbers, pH, and [Cl⁻] with a set of differential equations (Fig. 5 and Supplementary Fig. 8). Since CLC-type Cl⁻-H⁺ exchangers are absent in most synaptic vesicles[51], model vesicles contain V-type ATPases as the only source of acidification. The two transport modes of VGLUTs are described as two entities: a H⁺-glutamate exchanger/aspartate uniporter (VGLUT) and an additional chloride channel (VGLUT$_{Cl}$) (Fig. 5). Both processes depend on luminal pH with the same pK$_M$ (5.5) and on luminal [Cl⁻] with the same concentration dependence (Supplementary Fig. 5). Open probabilities of VGLUT Cl⁻ channels and maximum transport rates for VGLUT glutamate transporters were assumed to be voltage independent. Proton leak channels are required to describe constant steady-state vesicular pH during sustained V-type ATPase proton pumping. Cation transporter/channels were not included in the model.

Overall, the model consists of the following rate equations for luminal H⁺, pH, Cl⁻, and amino acid (AA) molecules:

$$\frac{d[AA^-]_L}{dt} = J_{VGLUT}, \tag{10}$$

$$\frac{d[H^+]_L}{dt} = J_{H^+} + J_{V_{ATPase}} - J_{VGLUTs}, \tag{11}$$

$$\frac{dpH_L}{dt} = -\frac{1}{N_A}\left(\frac{dH_L^+}{dt}\frac{1}{V}\right)\left(\frac{1}{\beta_{pH}}\right), \tag{12}$$

$$\frac{d[Cl^-]_L}{dt} = J_{VGLUT_{Cl}}. \tag{13}$$

Where $N_A$, $\beta_{pH}$ ($= \frac{40mM}{pH}$), and V represent Avogadro's number, the pH-buffering capacity of the vesicle, and the vesicle volume assuming a radius of 20 nm[52]. The equations for proton leak ($J_{H^+}$) and V-type ATPase ($J_{V_{ATPase}}$) are adapted from[32,33,53] and given as

$$J_{H^+} = P_{H^+} \times S \times \frac{\frac{\Delta\psi \times F}{R \times T}}{1 - e^{\frac{\Delta\psi \times F}{R \times T}}} \cdot \left(10^{-pH_C} \times e^{\frac{\Delta\psi \times F}{R \times T}} - 10^{-pH_{SV}}\right) \times \frac{N_A}{10^3}. \tag{14}$$

with $P_{H^+}$ (set to $\frac{1 \times 10^{-3} cm}{s}$) representing the permeability per unit area of proton leak, S the surface area of the vesicle, $\Delta\psi$ the membrane potential, $pH_C$ the cytosolic pH, $pH_{SV}$ the vesicular pH, and T room

temperature (resulting in $\frac{RT}{F} = 25.69$ mV), and

$$J_{V_{ATPase}} = <N_{VATP}> \times J_{VATP}(pH_L, \Delta\psi) \tag{15}$$

$J_{VATP}$ is the proton pumping rate of a single V-type ATPase complex as function of membrane potential and pH gradient. The proton pumping profile was generated using the model in[32,33,52] with $\Delta\psi$ varying from −200 to 500 mV and $pH_L$ varying from 1 to 14. During the simulation, pumping rates for current $\Delta\psi$ and $pH_L$ were retrieved from the profile and fed to the model at each given time step. In continuum lysosome models, the average number of V-type ATPase per lysosome ($<N_{VATP}>$) is usually assumed to be 300[32,33,53], however, recent high-resolution imaging experiments provided much lower numbers[54]. We assumed $<N_{VATP}> = 1$ for synaptic vesicles because of their much smaller size and chose the average number of VGLUTs per vesicle ($<N_{VGLUT}>$) = 1 as supported by functional and genetic data[55]. Supplementary Fig. 7 depicts predicted time constants of glutamate accumulation for various $<N_{VATP}>$ and $<N_{VGLUT}>$. The numbers provide average values and therefore also assume non-integer values in these simulations.

The following equations describe glutamate/H⁺ (Eq. 16) or aspartate fluxes (Eq. 17) through VGLUTs ($J_{VGLUT}$):

$$J_{VGLUT} = <N_{VGLUT}>V_{maxglu}f([H^+]_L)g([Cl^-]_L)\left(\Delta\psi - 0.5 \times \frac{RT}{F}\ln\frac{[glutamate]_L[H^+]_C}{[glutamate]_C[H^+]_L}\right) \tag{16}$$

$$J_{VGLUT} = <N_{VGLUT}>V_{maxasp}f([H^+]_L)g([Cl^-]_L)\left(\Delta\psi - \frac{RT}{F}\times\ln\frac{[asparate]_L}{[asparate]_C}\right). \tag{17}$$

where $V_{maxAA} = 100\,\text{s}^{-1}\text{mV}^{-1}$ is the maximum VGLUT glutamate or aspartate transport rate and $<N_{VGlut}>$ the number of vesicular glutamate transporters per vesicle. $f([H^+]_L)$ and $g([Cl^-]_L)$ are Michaelis–Menten relationships from fits to experimental data in Fig. 1d and Supplementary Fig. 4b, respectively. Cl⁻ fluxes through VGLUTs ($J_{VGLUT_{Cl}}$) are derived using an equation similar to Eq. 14 except that the reversal potential only depends on the Cl⁻ gradient.

$$J_{VGLUT_{Cl}} = <N_{VGLUT}>V_{maxCl}f([H^+]_L)g([Cl^-]_L)\left(\Delta\psi - \frac{RT}{F}\times\ln\frac{[Cl^-]_L}{[Cl^-]_C}\right) \tag{18}$$

$f([H^+]_L)$ and $g([Cl^-]_L)$ are again Michaelis–Menten relationships from fits to experimental data in Fig. 1b and Supplementary Fig. 4b, respectively, and $V_{maxCl}$ was set to 100 s⁻¹mV⁻¹. The equation for $\Delta\psi$ was modified from[33] to incorporate the effect of the negative charge on the AA, that is,

$$\Delta\psi = \frac{F\times V}{C}\left([H^+]_L + [K^+]_L + [Na^+]_L - [Cl^-]_L - [AA^-]_L - B\right) \tag{19}$$

with $C = C_0 \times S$ being the total capacitance of the vesicle with $C_0 = 1\,\mu\text{F/cm}^2$. $[K^+]_L$ and $[Na^+]_L$ were fixed to extracellular concentrations; i.e. to 5 mM and 145 mM. $B$ is the luminal concentration of impermeant charges given by the conservation of charge in the vesicle under initial conditions (indicated by the subscript 0).

$$B = [H^+]_{L,0} + [K^+]_{L,0} + [Na^+]_{L,0} - [Cl^-]_{L,0} - [AA^-]_{L,0} - \frac{C}{F\times V}(\psi_{in} - \psi_{out}) \tag{20}$$

Finally, the effect of surface charge on various ion concentrations at the membrane was also taken into account (as shown in[33]), with the inner ($\psi_{in}$) and outer ($\psi_{out}$) leaflets potentials set to 0 mV and −50 mV, respectively.

Starting conditions for the mathematical modeling reflect the ion concentrations immediately after endocytosis of synaptic vesicles. The vesicular lumen contained solutions resembling the external solutions, with high [Cl⁻], neutral pH and negligible glutamate/aspartate concentrations; cytoplasmic [Cl⁻] ($Cl^-_C$) was set to 21 mM (as obtained from gramicidin-perforated patch recordings on the Calyx of Held[56]) and cytoplasmic [glutamate] and [aspartate] to 10 mM[57].

### Confocal imaging
Confocal imaging of living HEK293T cells bathed in standard external solution was conducted using an inverted confocal laser scanning microscope (Leica TCS SP5, Leica Microsystems, Heidelberg, Germany) with a 63×/1.4 oil immersion objective. GFP was excited using a 488 nm laser and fluorescence emission was detected between 490 and 580 nm.

### Quantification and statistical analysis
Data are given either as the mean ± 95% confidence interval (CI) or as boxplots, including all single data and representing the mean value, median, upper and lower quartiles, and 95% CI. Measurements were taken from distinct samples. Mean and CIs of fit parameters and fitted curves are either from individual fits (single-channel currents or time-constants) or determined by bootstrap analysis[58] (slope of $E_{rev} - E_H$ plots, glutamate transport rates). For patch clamp electrophysiology experiments, sample sizes correspond to the number of patched cells and are given in the figure or legend. For statistical analysis of two groups, we used two-tailed $t$-test for normally distributed data and the Mann–Whitney $U$-test for non-normally distributed data.

### Reporting summary

Further information on research design is available in the Nature Portfolio Reporting Summary linked to this article.

## Data availability
The data that support this study are available from the corresponding authors upon request. The source data underlying Figs. 1–3, 5, 6 and Supplementary Fig. 1–7 are provided as a Source Data file. Source data are provided with this paper at [https://jugit.fz-juelich.de/computational-neurophysiology/variable-coupling-vglut]. Source data are provided with this paper.

## Code availability
Source code of the kinetic model for glutamate accumulation in synaptic vesicles is provided at [https://jugit.fz-juelich.de/computational-neurophysiology/variable-coupling-vglut].

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

## Acknowledgements

We thank Yannick Güthoff for performing the tail current analysis in Fig. 2f, Dr. Stefanie Bungert-Plümke for performing the biochemical analysis of H191K-H426K-D428Q VGLUT1$_{PM}$ in Supplementary Fig. 2i, and A. Franzen and P. Thelen for their excellent technical support. This work was supported by the Deutsche Forschungsgemeinschaft (German Research Foundation) to Ch.F. (FA 301/15–2) as part of Research Unit FOR 2518, *DynIon*; to Ch.F. and G.U. as part of the Research Unit FOR 2795, to REG (GU 2042/2-1), and by the NIH to G.U. (R01 AG053988).

## Author contributions

B.K. designed and performed most of the experiments and generated most of the data and figures for the manuscript; B.B. performed experiments on WT and H120A H⁺-glutamate coupling stoichiometry and the majority of the control experiments; D.K. designed and evaluated most of the noise analysis experiments; V.L. performed experiments on WT and H120A H⁺-glutamate coupling stoichiometry; C.H. and R.E.G. engineered the VGLUT1PM variant; G.U. generated the quantitative synaptic vesicle model; and Ch.F. conceived the idea, designed experiments, supervised the work, and wrote the manuscript.

## Funding

## Competing interests

The authors have no competing interests.
