## [Peer Review File · Nature Communications]

Vesicular glutamate transporters are H⁺-anion exchangers that operate at variable stoichiometryReviewers' Comments:

Reviewer #1:

Remarks to the Author:

This work takes advantage of a plasma membrane-targeted version of vesicular glutamate transporter VGLUT1 to record the associated currents and address a long-standing question about whether the VGLUTs operate as H⁺ exchangers, like the other neurotransmitter transporters on synaptic vesicles (SVs). Most previous work has suggested that the VGLUTs do not exchange H⁺ for glutamate, and dissipation of the pH gradient across the SV membrane can increase glutamate uptake, but luminal pH affects SV membrane potential, so these variables are not independent. In this work, the authors use the gold standard of a shift in reversal potential to deduce a difference between the flux of glutamate from other anions that suggests H⁺ exchange. This effectively excludes kinetic effects such as the allosteric activation by protons which the authors also confirm here does occur.

The problem is that the authors show essentially no controls to document that the currents derive specifically from VGLUT1. They define the currents by activation at low external pH, and this resembles previous work, but many currents will also be affected by pH. Since the permeant anions (inorganic and organic) are loaded into the cell, it is also not possible to define the currents using them. The authors repeatedly state that the background currents are low but show only a single panel documenting the dependence of these currents on transfection of VGLUT1. In the text, they mention currents of similar size for control and transfected HEK cells at pH 7.4 but since there are no VGLUT currents at pH 7.4, this does not mean much. Only in Figure 3 do they show a control cell with low currents, but these are background-subtracted and it would be important to see what the endogenous currents look like so that readers can evaluate the data. In addition to providing these essential controls for the experiments involving Cl⁻ and glutamate, the authors must show similar controls for the other anions used, as these may each have their own, different background currents and the authors do not seem to have considered this. The reason why this is so important is that although some of the findings fit previous work, others do not and still others do not make sense. Although the dependence on external/luminal pH is as expected, the permeation of multiple inorganic anions is not consistent with multiple radiotracer flux and electrophysiologic studies. Also, the permeation of large organic anions, although in principle possible, is very difficult to reconcile with the reported specificity of the VGLUTs. The authors try to reconcile this by suggesting that lack of coupling to protons accounts for the failure of previous work to detect this flux, but aspartate is barely recognized even as an inhibitor of the VGLUTs, and this property should not be affected by proton coupling. Since controls are essential to interpret all of the data, the authors should also use a nonfunctional mutant and one of the known inhibitors.

With regard to the shifts in reversal potential, these are in the right direction and probably of the right magnitude for proton exchange, but the absolute values are very difficult to understand—they should all be positive, not negative. In addition, it is hard to understand how an organic anion only in the cytoplasm yields outward currents if the external Cl⁻ cannot enter due to strong rectification of that conductance. Since reversal potential depends on these outward currents and it is not clear how they originate, it is very difficult to use them to infer H⁺ coupling by the VGLUTs. The results with His120 are interesting and the comparison of wild-type and mutant at least provides some kind of control.

In summary, the authors address an interesting question and some of the data look reasonable, but others do not and the lack of appropriate controls is egregious, particularly when they draw far-reaching conclusions. It may be that some of these currents reflect VGLUT activity (the Cl⁻ currents, for example) but others (the currents due to organic anions) do not. To draw conclusions about the VGLUTs, the authors must therefore demonstrate that the effects they observe reflect the activity of VGLUT1, and the magnitude of the signal will help to evaluate the results. The idea that protons serve different roles for Cl⁻ conductance and glutamate transport is interesting, but the modeling does not add much and remains highly speculative, particularly given the uncertainty about the experimental results.

Reviewer #2:

Remarks to the Author:

This report details the electrophysiological characterization of VGLUT1, a vesicular glutamate transporter. While VGLUTs are normally localized to secretory vesicles, the authors used a mutated transporter, which does not completely internalize, and, thus, can be studied with regular patch clamp recording from the plasma membrane. This "trick" has been used before by other authors.

Interestingly, a large, proton dependent anion channel is observed, which is measured by current recording of anion outflow from whole cells. The authors also measure glutamate-induced currents, which are much smaller, presumably due to the active transport of glutamate across the membrane (actual flux has not been measured). Furthermore, noise analysis is performed, allowing the determination of unitary conductance and channel open probability of the chloride channel. This latter part is novel and provides previously unavailable information about VGLUTs. Finally, the authors propose a proton counter-transport mechanism of glutamate transport, based on reversal potential measurements. This aspect is interesting, although previous results on VGLUT2 (Eriksen et al. Neuron, 2016) did not identify proton exchange, but rather a regulatory mechanism for the proton effect. It is surprising that this report is not discussed in more detail.

Overall, this is an interesting study, but some major and minor issues were identified:

Major points:

The reversal potential of VGLUT2 currents was not dependent on external pH, speaking against a coupling of protons to glutamate transport (Eriksen et al. Neuron, 2016). How can this finding be reconciled with the conflicting findings in this report?

It is surprising that VGLUT inhibitors were not used to determine specificity of the currents. It would be useful to measure currents in the presence of nM affinity inhibitors, such as, for example, Trypan Blue.

Along the same lines, Cl⁻ channel inhibitors, such as DIDS, were used in other reports to block the VGLUT chloride conductance. Why were such control experiments not done?

A glutamate dose response curve needs to be shown to test if the K_m is in line with previous estimates.

The kinetic model used to calculate vesicular glutamate loading is not shown. This has to be described in much more detail, in particular with respect to the important parameters, such as transport rates, numbers of V-ATPase and VGLUTs in the membrane etc.

As to the previous point, it is surprising that the modeled time course for vesicle loading is so slow. Should the x-axis be in ms, rather than s? Considering the high frequency firing capability of glutamatergic synapses, it would seem that such slow vesicle loading would present a rate limiting step?

Page 7: Was fluorescence measured from the whole cell, or only from the plasma membrane? If the former is the case, how does transporter in intracellular compartments affect the outcome of the fluorescence quantification?

Minor points:

Does bicarbonate compete with glutamate for the transporter binding site? Since bicarbonate is a major physiological anion present at high concentrations, this is a relevant question.

Does VGLUT1 WT elicit membrane currents?

Page 4: What is an "inward holding current" ? Please elaborate.

What is the "late" current? Please define.

Why do equations 4 and 5 not contain the glutamate concentration?

Responses to reviewers' comments

Reviewer #1: *This work takes advantage of a plasma membrane-targeted version of vesicular glutamate transporter VGLUT1 to record the associated currents and address a long-standing question about whether the VGLUTs operate as H⁺ exchangers, like the other neurotransmitter transporters on synaptic vesicles (SVs). Most previous work has suggested that the VGLUTs do not exchange H⁺ for glutamate, and dissipation of the pH gradient across the SV membrane can increase glutamate uptake, but luminal pH affects SV membrane potential, so these variables are not independent. In this work, the authors use the gold standard of a shift in reversal potential to deduce a difference between the flux of glutamate from other anions that suggests H⁺ exchange. This effectively excludes kinetic effects such as the allosteric activation by protons which the authors also confirm here does occur.*

“ 1. The problem is that the authors show essentially no controls to document that the currents derive specifically from VGLUT1. They define the currents by activation at low external pH, and this resembles previous work, but many currents will also be affected by pH. Since the permeant anions (inorganic and organic) are loaded into the cell, it is also not possible to define the currents using them. The authors repeatedly state that the background currents are low but show only a single panel documenting the dependence of these currents on transfection of VGLUT1. In the text, they mention currents of similar size for control and transfected HEK cells at pH 7.4 but since there are no VGLUT currents at pH 7.4, this does not mean much. Only in Figure 3 do they show a control cell with low currents, but these are background-subtracted and it would be important to see what the endogenous currents look like so that readers can evaluate the data. In addition to providing these essential controls for the experiments involving Cl⁻ and glutamate, the authors must show similar controls for the other anions used, as these may each have their own, different background currents and the authors do not seem to have considered this. The reason why this is so important is that although some of the findings fit previous work, others do not and still others do not make sense.”

We would like to thank reviewer 1 for pointing out the necessity of showing additional control experiments. Many of these results were already existing, and we apologize for not having shown them in the original version. We now provide the results of three different types of control experiments, the comparison of currents in transfected and non-transfected cells (new Supplementary Fig. 2 a-d), the correlation of VGLUT1_{PM} expression levels and anion currents (revised Figs. 1a and 2i) as well as block by an established VGLUT1 blocker, Rose Bengal (new Supplementary Fig. 2 e,f).

New supplementary Figs. 2 a-d compare currents from untransfected cells with cells expressing WT VGLUT1_{PM} for all tested anions. These experiments were performed during external perfusion with standard external solution at pH 5.5 or pH 7.4; cells were dialyzed with internal solutions containing Cl⁻, Br⁻, I⁻, NO₃⁻, HCO₃⁻ (new Supplementary Fig. 2a and b), glutamate, aspartate, gluconate, MES, MSA, or isethionate (new Supplementary Fig. 2c and d). For small anions, background currents measure less than 3% of the VGLUT1_{PM} currents (new Supplementary Fig. 2a), and for large and polyatomic anions less than 15 % (new Supplementary Fig. 2c). Background currents are not activated by acidic external pH, and our correction protocol, i.e. subtracting currents measured at external pH 7.4 from results at acidic pH_o, permits an almost perfect correction of all background currents (relative corrected background currents for small and large/polyatomic anions below 3% of VGLUT1_{PM} currents,

Supplementary Fig. 2b and d). Especially, subtracted glutamate and aspartate background currents are close to 0 over the whole voltage range (Supplementary Fig. 2d).

Revised Figs. 1b and 2i depict plots of unsubtracting VGLUT1_{PM} currents versus whole-cell GFP fluorescences for small anions such as Cl⁻, Br⁻, I⁻ and NO₃⁻ (new Fig. 1b, Fig. 2a of the original manuscript) or for various large polyatomic internal anions (Fig. 2i). We routinely express VGLUT1_{PM} as GFP fusion protein, and whole-cell GFP fluorescences can thus be used to quantify the number of transporters in the whole cell. In such plots, we do not distinguish between transporters in different cell compartments (see also our response to point 4. of reviewer 2). For each anion, we observed a linear relationship between whole-cell currents at -160 mV and whole-cell fluorescences of individual cells under activating acidic conditions (pH 5.5). For all anions, currents at pH 7.4 were independent of VGLUT1_{PM} expression levels (data are given by small symbols in the same color as data at pH_{ext} = 5.5). These data exclude the possibility that the observed currents are mediated by endogenous proteins upregulated by expression of VGLUT1_{PM}, they fully support the notion that anion currents are mediated by heterologously expressed VGLUT1_{PM}. The linear relationship between currents and fluorescence demonstrates that current/fluorescence ratios report mean currents mediated by individual transporters.

New supplementary Figs. 2e and f illustrate block of VGLUT1_{PM} currents by Rose Bengal. Rose Bengal is a noncompetitive VGLUT blocker with high affinity (Ogita *et al.* (2001). *J Neurochem* **77**, 34-42; Pietrancosta *et al.* (2020). *Mol Neurobiol* **57**, 3118-3142). We tested the effect of various [Rose Bengal] on Cl⁻, NO₃⁻, HCO₃⁻, glutamate, aspartate, gluconate and MES currents; we did not test block of Br⁻, I⁻, MSA or isethionate currents because there are no indications for unique transport mechanisms for these anions. We found all anion currents to be blocked by Rose Bengal. Supplementary Fig. 2e compares currents from transfected cells at neutral pH with currents at pH 5.5 and 1600 nM Rose Bengal. With the exception of NO₃⁻, all anion currents are blocked to similarly small amplitudes at saturating [Rose Bengal]. For all anions, block was not complete, i.e. currents at saturating [Rose Bengal] were slightly larger than background currents. NO₃⁻ background currents are slightly larger than currents conducted by other anions (also visible in new Supplementary Fig. 2a and b); however, we believe that the very large VGLUT1_{PM} NO₃⁻ currents compensate for this larger unspecific current component. New supplementary Fig. 2f provides dose-response curves and fits with a Michaelis-Menten relationship. These data suggest that VGLUT1_{PM}-mediated Cl⁻ and NO₃⁻ currents are more effectively blocked by Rose Bengal than currents carried by larger anions. Preliminary experimental data revealed different K_{on} and K_{off} reaction rates for Rose Bengal block of Cl⁻ and glutamate. We are planning to study these processes in more detail in the near future, however, we feel that the kinetic of Rose Bengal block are out of the scope of this manuscript. The Rose Bengal data provide additional support that, for all anions, whole-cell currents in transfected HEK293T cells are mediated by VGLUT1_{PM}.

We now discuss these results on p.4, lines 11 to p. 5, line 6, and on p.8, line 14 to p.9, line 2. These data, taken together, virtually exclude the possibility that endogenous currents significantly contribute to any of the current components shown in our manuscript. They strongly support the notion that VGLUT1_{PM} transports not only glutamate, but also multiple other large anions.

“2. Although the dependence on external/luminal pH is as expected, the permeation of multiple inorganic anions is not consistent with multiple radiotracer flux and electrophysiologic studies.

Also, the permeation of large organic anions, although in principle possible, is very difficult to reconcile with the reported specificity of the VGLUTs. The authors try to reconcile this by suggesting that lack of coupling to protons accounts for the failure of previous work to detect this flux, but aspartate is barely recognized even as an inhibitor of the VGLUTs, and this property should not be affected by proton coupling. “

We agree with reviewer 1 that some of our results are unexpected. However, they do not contradict published earlier results. Cl⁻, Br⁻, I⁻, glutamate⁻ and aspartate⁻ have been used as anions in earlier electrophysiological studies of VGLUT, and aspartate currents have been already in endosome patch clamp recordings (Fig. 1 from Chang et al. (2018). *Elife* **7**, e34896, doi: 10.7554/eLife.348962018), with shifted reversal potential, in full agreement with our results. Other anions, such as gluconate, MES, MSA or isethionate, have not been tested in any published report. The lower driving force for aspartate uptake together with the pronounced rectification predicts negligible aspartate uptake in radioactive studies.

Aspartate fails to reduce ³H-glutamate uptake, in contrast to L-glutamate and D-glutamate (Moriyama and Yamamoto (1995) *J Biol Chem* **270**, 22314-22320; Bellocchio *et al.* (2000). *Science* **289**, 957-960). In these experiments, aspartate was added either at a concentration of 1 mM (Moriyama and Yamamoto (1995) *J Biol Chem* **270**, 22314-22320) or 10 mM (Bellocchio *et al.* (2000). *Science* **289**, 957-960). This result is usually interpreted as absence of aspartate transport, in full agreement with experiments on the amino acid content of synaptic vesicles (Burger et al. (1989). *Neuron* **3**, 715-720) and the results of our continuum simulations. Since L-glutamate binds with an apparent affinity in the mM range, these findings only indicate that aspartate does not bind tighter to the transporter than glutamate.

There is experimental support that aspartate can bind to the transporter: aspartate has been reported to block VGLUT1 Cl⁻ currents (Eriksen *et al.* (2016). *Neuron* **90**, 768-780, Fig. 3d). The same publication also shows block of VGLUT1 Cl⁻ currents by glutamate, demonstrating that even the transport substrate represents only a very poor blocker. We now cite this work on p. 18, line 22 of the revised manuscript. We furthermore now cite earlier work on the specificity of VGLUTs on p. 8, line 7.

“3. Since controls are essential to interpret all of the data, the authors should also use a nonfunctional mutant and one of the known inhibitors.”

As outlined in our response to comment 1., we now show block of VGLUT1_{PM} currents by Rose Bengal for the majority of the tested anions. In attempts to identify a nonfunctional mutant, we studied various mutant transporters that were reported to be non-functional, such as R80A or R314A VGLUT1_{PM}, and could also record currents for these mutants that exceeded background currents for certain anions. Despite extensive mutagenesis, we have not identified any mutant VGLUT1_{PM} that is completely non-functional with internal NO₃⁻.

We therefore used an alternative experiment to test whether the observed currents are carried by endogenous transporters that are upregulated by heterologous overexpression of VGLUT1_{PM}. We studied the correlation between transporter expression and anion current for all anions (revised Figs. 1b and 2i). We feel that the linear relationship of these two parameters together with the effective block of VGLUT1_{PM} currents by Rose Bengal suffices to exclude this possibility. We discuss these issues on p.5, line 1 to line 6, and on p. 8, line 14 – line 25.

“4. With regard to the shifts in reversal potential, these are in the right direction and probably of the right magnitude for proton exchange, but the absolute values are very difficult to

understand—they should all be positive, not negative. In addition, it is hard to understand how an organic anion only in the cytoplasm yields outward currents if the external Cl⁻ cannot enter due to strong rectification of that conductance. Since reversal potential depends on these outward currents and it is not clear how they originate, it is very difficult to use them to infer H⁺ coupling by the VGLUTs. The results with His120 are interesting and the comparison of wild-type and mutant at least provides some kind of control.”

Rectification of VGLUT1_{PM} Cl⁻ currents is caused by voltage-dependent gating (Fig. 1i). This rectification is present in the presence of small anions such as Cl⁻, Br⁻ or I⁻ in the intracellular solution (Fig. 1). Intracellular dialysis with polyatomic anions such as glutamate, aspartate or gluconate modifies gating of VGLUT1_{PM} anion channels. In cells dialyzed with solutions containing polyatomic anions, deactivation of VGLUT1_{PM} currents is incomplete, resulting in open channels/active transporters also at positive potentials (Fig. 2). We now explicitly refer to the altered gating behavior on p. 9, line 3. Additionally, we are now addressing this modulation of VGLUT1_{PM} currents in a new Supplementary Fig. 5 that illustrates how internal glutamate prevents VGLUT1_{PM} current deactivation and thus permits reversal potential measurements at positive potentials with internal glutamate, aspartate or gluconate. We describe this important finding on p. 9, line 3 to line 17 of the revised manuscript.

In the original version of our manuscript, we assessed H⁺-glutamate coupling by studying whether variation of internal pH is associated with shifts of the current reversal potentials. We performed such analysis at five internal pH values for two different external glutamate concentrations. In these experiments, polyatomic anions are on both membrane sites, and glutamate transport can thus occur in both directions. We observed reversal potentials that changed linearly with the Nernst potential for H⁺ (E_H) with a slope close to 0.5. Such slope is expected for a 1:1 H⁺-glutamate exchanger. Reviewer 1 is completely right, the experimentally observed reversal potentials are more negative than expected for this transport stoichiometry. The reason for this difference between theoretical and experimentally observed reversal potentials is the presence of Cl⁻ in the external solution and the dual function of VGLUTs as glutamate transporter and Cl⁻ channel. External Cl⁻ is obligatorily required, without Cl⁻ VGLUT1_{PM} cannot function as glutamate transporter (Supplementary Fig. 4). We performed many experiments with [Cl⁻]_o = 40 mM; at this concentration, Cl⁻ inward fluxes at positive potentials are small, and glutamate transport is still maximally activated (Supplementary Fig. 4a). However, even under these conditions, Cl⁻ influx is expected to make experimentally determined current reversal potentials more negative than theoretical values of H⁺-glutamate exchangers.

In the revised version we now first show pH_i-dependent shifts in E_{rev}s of glutamate currents in the new Fig. 3a. Moreover, we report that VGLUT1_{PM} current reversal potentials at pH_i = 8.5 are positive to values expected for a glutamate uniporter, even in presence of hyperpolarizing VGLUT1_{PM}-mediated Cl⁻ influx. These data are incompatible with glutamate uniport, but with transport coupling to choline. A positive E_{rev} under our conditions might be due to additional Cl⁻-glutamate exchange, however such coupling can be excluded by the Cl⁻ dependence of E_{rev}s (supplementary Fig. 6a). Absolute values of E_{rev} and their pH_i dependence thus indicate that VGLUT1_{PM} functions as H⁺-glutamate exchanger (p. 10, line 4 to 22).

To assess how VGLUT1_{PM} Cl⁻ currents affect the coupling stoichiometries obtained from E_{rev}-E_H plots we measured VGLUT1_{PM} currents in cells dialyzed with glutamate-based solution at three consecutively applied external [Cl⁻] (20, 40 and 100 mM) and performed such experiments

with three distinct internal pH. Lower $[Cl^-]_o$ (10 mM) resulted in significantly reduced glutamate currents, which were too small for accurate reversal potential measurements. New Fig. 3b provides a plot of the current reversal potential versus the ratio of current amplitudes at +60 mV by the current amplitude at -140 mV for two different pH_i and three $[Cl^-]_o$. Since cells are dialyzed with glutamate as only anion, these current ratios provide relative Cl^- current amplitudes. Our data demonstrate that, at a given relative Cl^- current amplitude, E_{revS} assumes different values for $pH_i = 8.5$ and $pH_i = 6.5$. This demonstrates that pH_i -dependent changes of E_{rev} are caused by altered driving forces for glutamate transport and not by changes in Cl^- channel activity (p. 11, line 11 – p. 12, line 9). A plot of E_{rev} versus relative Cl^- currents for aspartate currents demonstrate that aspartate currents reverse at more negative potentials than glutamate currents at comparable Cl^- current amplitudes, indicating smaller driving forces for aspartate than for glutamate (Fig. 3c) and partial uncoupling of H^+ -driven transport by aspartate (p. 11, line 22 – 25).

Fig. 3d shows $E_{rev}-E_H$ plots at three external $[Cl^-]_o$, showing that VGLUT1_{PM} Cl^- currents affect the obtained E_{revS} and reduce the slope of the $E_{rev}-E_H$ relationship, as every leakage current in transport stoichiometry assessment via reversal potential measurements. Consecutive perfusion with the different external $[Cl^-]$ results in higher leakage currents for a subset of experiments. Therefore, slope factors of the $E_{rev}-E_H$ plots are slightly lower than in the experiments shown in Fig. 3a, which we use exclusively for estimating the transport stoichiometry.

We now discuss the novel results on p. 17, line 20 – p. 18, line 5, stating that our results are in agreement with a 1:1 H^+ -glutamate exchange, but that we cannot exclude higher coupling ratios.

Chang et al. (2018) (*Elife* 7, e34896, doi: 10.7554/eLife.348962018) reported the Cl^- independent mutant R176A VGLUT1_{PM}. With these mutant transporters, glutamate currents can be measured in complete absence of Cl^- and provide biionic reversal potentials. We performed experiments with R176A VGLUT1_{PM}. Unfortunately, the mutation reduces current amplitudes in our hands and prevents conclusive insights into transport coupling. Moreover, such experiments will only provide insights into the mutant, but not in WT transporters. We therefore did not include such experiments into our revised manuscript.

“5. In summary, the authors address an interesting question and some of the data look reasonable, but others do not and the lack of appropriate controls is egregious, particularly when they draw far-reaching conclusions. It may be that some of these currents reflect VGLUT activity (the Cl^- currents, for example) but others (the currents due to organic anions) do not. To draw conclusions about the VGLUTs, the authors must therefore demonstrate that the effects they observe reflect the activity of VGLUT1, and the magnitude of the signal will help to evaluate the results. The idea that protons serve different roles for Cl^- conductance and glutamate transport is interesting, but the modeling does not add much and remains highly speculative, particularly given the uncertainty about the experimental results.”

We would like to thank reviewer 1 for his rigorous and critical review. We apologize for not having provided sufficient controls already with the original version of our submission; many of these results were existing, and we would not have based far-reaching conclusions on our data without such controls. We feel that the controls we are now showing in the revised Figs. 1 and 2 and the new Supplementary Fig. 2 remove remaining doubts about the specificity of the measured currents. We also would like to thank for the critical analysis of the reversal potential measurements. We feel that the additional analysis of possible reasons for the shifts in reversal potentials substantially strengthen the experimental support for variable H^+ coupling in

vesicular glutamate transporters. After substantiating this particular functional feature, we believe that our continuum model of synaptic neurotransmitter accumulation provides important insights into the consequence of variable H^+ coupling for glutamate accumulation in synaptic vesicles.

Reviewer #2: *This report details the electrophysiological characterization of VGLUT1, a vesicular glutamate transporter. While VGLUTs are normally localized to secretory vesicles, the authors used a mutated transporter, which does not completely internalize, and, thus, can be studied with regular patch clamp recording from the plasma membrane. This "trick" has been used before by other authors. Interestingly, a large, proton dependent anion channel is observed, which is measured by current recording of anion outflow from whole cells. The authors also measure glutamate-induced currents, which are much smaller, presumably due to the active transport of glutamate across the membrane (actual flux has not been measured). Furthermore, noise analysis is performed, allowing the determination of unitary conductance and channel open probability of the chloride channel. This latter part is novel and provides previously unavailable information about VGLUTs. Finally, the authors propose a proton counter-transport mechanism of glutamate transport, based on reversal potential measurements. This aspect is interesting, although previous results on VGLUT2 (Eriksen et al. Neuron, 2016) did not identify proton exchange, but rather a regulatory mechanism for the proton effect. It is surprising that this report is not discussed in more detail.*

Overall, this is an interesting study, but some major and minor issues were identified:

Major points:

"1. The reversal potential of VGLUT2 currents was not dependent on external pH, speaking against a coupling of protons to glutamate transport (Eriksen et al. Neuron, 2016). How can this finding be reconciled with the conflicting findings in this report?"

Eriksen *et al.* performed experiments on intact oocytes. Although oocytes contain glutamate, whole-oocyte currents are predominantly Cl^- currents mediated by VGLUT1_{PM} chloride channels, and no conclusions were made from these experiments about VGLUT glutamate currents. The authors only used the pH dependence of current reversal potentials to obtain insights into possible proton coupling of the Cl^- currents and concluded that " *H^+ activates without contributing to the currents observed*", in full agreement with our results on the channel-like conduction of small anions by VGLUT1_{PM}. We now discuss the importance of the result for VGLUT1 glutamate transport on p. 17, line 4 and for VGLUT Cl^- conduction on p. 16, line 5.

"2. It is surprising that VGLUT inhibitors were not used to determine specificity of the currents. It would be useful to measure currents in the presence of nM affinity inhibitors, such as, for example, Trypan Blue. Along the same lines, Cl^- channel inhibitors, such as DIDS, were used in other reports to block the VGLUT chloride conductance. Why were such control experiments not done?"

We now test the effect of Rose Bengal on VGLUT1_{PM} currents (supplementary Fig. 2e and f, p. 8, line 12 - 17). Rose Bengal is a noncompetitive VGLUT blocker with high affinity (Ogita *et al.* (2001). *J Neurochem* **77**, 34-42; Pietrancosta *et al.* (2020). *Mol Neurobiol* **57**, 3118-3142). The effect of various [Rose Bengal] on Cl^- , NO_3^- , HCO_3^- , glutamate, aspartate, gluconate, and MES currents were tested; we did not test block of Br^- , I^- . MSA or isethionate currents because

there are no indications for unique transport mechanisms for these anions. Supplementary Fig. 2e compares currents from transfected cells at neutral pH with currents at pH 5.5 and 1600 nM Rose Bengal. With the exception of NO_3^- , all anion currents are blocked to similarly small amplitudes at saturating [Rose Bengal]. For all anions, block was not complete, i.e. currents at saturating [Rose Bengal] were slightly larger than background currents. NO_3^- background currents are slightly larger than currents conducted by other anions (also visible in new Supplementary Fig. 2a and b); however, we believe that the very large VGLUT1_{PM} NO_3^- currents compensate for this larger unspecific current component.

Rose Bengal blocks VGLUT1_{PM} carried by small anions with lower K_{MS} than VGLUT1_{PM}-mediated transport of polyatomic anions (supplementary Fig. 2 F). We did some initial analyses on Rose Bengal blocking/unblocking time courses for VGLUT1_{PM} Cl^- and glutamate⁻ currents. Cytoplasmic glutamate dramatically reduces on- as well as off-rates of block by Rose Bengal. We plan to further study these interesting findings in the near future, but feel that such experiments are out of the scope of the current manuscript. To not further extend the time required for this revision, we decided against additional testing of other VGLUT blockers, such as Trypan Blue.

DIDS is quite unspecific and blocks not only VGLUTs, but also CLC anion channels and transporters and various other anion transporters (Cabantchik and Greger (1992). *Am J Physiol* **262**, C803-827; Howery *et al.* (2012). *Chem Biol* **19**, 1460-1470). Moreover, it is instable and often converted into polythiourea compounds with different blocking potencies (Matulef *et al.* (2008). *ACS Chem Biol* **3**, 419-428). Although DIDS block substantially contributed to assigning VGLUT1 Cl^- currents (Eriksen *et al.* (2016). *Neuron* **90**, 768-780), these features restrict the interpretability of blocking experiments with small background currents. We therefore refrained from using DIDS for control experiments.

“3. A glutamate dose response curve needs to be shown to test if the K_m is in line with previous estimates.”

Since we are studying currents rather than transport of radioactive substrates, such glutamate dose responses would require substitution of glutamate with a non-transported/impermeant anion. Unfortunately, we have not found any anion with such properties. Moreover, because of its pronounced rectification we can only record VGLUT1_{PM} glutamate currents out of the cell with the experimental setup of our study, making substitution experiments even more difficult. We can therefore not provide glutamate dose responses. We measured pH and Cl^- dose response curves for glutamate and Cl^- currents and observed good agreement with published data.

“4. The kinetic model used to calculate vesicular glutamate loading is not shown. This has to be described in much more detail, in particular with respect to the important parameters, such as transport rates, numbers of V-ATPase and VGLUTs in the membrane etc.”

We extended and improved the description of the kinetic model (p. 25, line 9 to p. 28, line 13), paying special attention that these numbers are easily accessible to the future readers. The equations for all fluxes and the parameters used are now described in detail. We also added a schematic diagram as inset to Fig. 4 visualizing the main pathways considered in the model.

“5. As to the previous point, it is surprising that the modeled time course for vesicle loading is so slow. Should the x-axis be in ms, rather than s? Considering the high frequency firing capability of glutamatergic synapses, it would seem that such slow vesicle loading would present a rate limiting step?”

We would like to thank the reviewer for bringing up this important point. Hori and Takahashi studied the filling of synaptic vesicle with glutamate in slices of Calyx of Held by estimating the time course of miniature EPSP recovery after uncaging of L-glutamate (Hori and Takahashi (2012). *Neuron* **76**, 511-517) and obtained a time constant ranging from 10 to 50 seconds at room temperature and upon uncaging 0.5 mM to 2 mM L-glutamate. In cultured hippocampal neurons, Herman et al. ((2018). *Frontiers Synapt Neurosci* **10**, 44) obtained slower time constants and concluded that there is variability in the filling time course filling between different synaptic vesicles. These experimental data indicate that vesicle filling is a indeed a slow process occurring on time scales of seconds.

At the default parameters used, our model results in a time constant of 36 s. We now tested how changing the average number of VGLUTs or of VATPase complexes per vesicle modifies the time constant for vesicle filling in our model. We discuss this question on p.14, line 3 and add a new Supplementary Fig. 7.

“6. Page 7: Was fluorescence measured from the whole cell, or only from the plasma membrane? If the former is the case, how does transporter in intracellular compartments affect the outcome of the fluorescence quantification?”

We measure whole-cell fluorescences for every cell prior to seal formation. This value is proportional to the number of transporters per cell and does not distinguish between intracellular compartments and the surface membrane. However, since we are imaging every cell under identical conditions, the percentage of transporters in the surface membrane will not be affected by subsequent internal dialysis or external perfusion. We therefore expect that a fixed percentage of all transporters is in the surface membrane, the whole-cell fluorescence can be used as value proportional to the number of transporters in the surface membrane. We cannot directly determine the number of transporters in the surface membrane from these values, but we can use it to normalize transporter currents by transporter expression levels.

We have used this approach for a variety of ion channels and transporters (Schänzler and Fahlke (2012). *J Physiol* **590**, 259-272; Ronstedt *et al.* (2015). *Sci Rep* **5**, 15382; Tan *et al.* (2017). *Front Physiol* **8**, 269; Guzman *et al.* (2022). *Front Mol Neurosci* **15**, 872407; Kovermann *et al.* (2022). *Epilepsia* **63**, 388-401). A possible limitation is the saturation of the intracellular trafficking machinery, which would cause a deviation from linearity for plots of the number of transporters in the surface membrane versus the number of transporters in the whole cell. For VGLUT1_{PM}, we see nicely linear relationships between whole-cell currents and fluorescence (Figs. 1B and 2I), in agreement with the assumptions of this approach. Work with the intracellular Cl⁻/H⁺ exchanger CIC-4 demonstrated that for transporters that mostly reside in the intracellular compartments (with only a minority of transporters in the surface membrane) ratios of currents and fluorescence are much less defined and that sequence alterations that improve surface membrane insertion greatly reduce the scatter around the linear fits (Guzman *et al.* (2022). *Front Mol Neurosci* **15**, 87240). VGLUT1_{PM} is mostly in the surface membrane, and fluorescence current plots exhibit much smaller scatter than for CIC-4. We now discuss possible limitations of this approach in more detail on p. 22, line 24 to p. 23, line 13.

We refrained from determining absolute numbers of transporters in the surface membrane; optical methods are prone to errors because separation of proteins in or close to the surface membrane is very difficult, and biochemical methods impossible in living cells prior to patch clamp. We felt that such experiments with cells after patch clamp might bear the risk of un-

accuracies because of protein redistribution during long-lasting perfusions with non-physiological solutions.

Minor points:

“1. Does bicarbonate compete with glutamate for the transporter binding site? Since bicarbonate is a major physiological anion present at high concentrations, this is a relevant question.”

The glutamate binding site has been recently defined by Cryo-EM (Li et al (2022) Allosteric regulation of a synaptic vesicle glutamate transporter. *bioRxiv* doi: 10.1101/2022.07.26.501550v1). In atomistic MD simulations, we observed coordination of glutamate by two arginine side chain at this site. In contrast, HCO_3^- only transiently binds to one of the two side chains, quite similarly as Cl^- . Since we have no indications for competition between Cl^- and glutamate (see supplementary Fig. 4B), we believe that there is also no competition between HCO_3^- and glutamate.

“2. Does VGLUT1 WT elicit membrane currents?”

Cells expressing WT VGLUT1 only exhibit currents slightly above background currents. We give this information now on p. 4, line 17 - 20: “In cells transfected expressing WT VGLUT1 without improved plasma membrane targeting, currents were much smaller (at -160 mV, $\text{pH}_o = 5.5$; 47 ± 27 pA, $n = 10$) than in cells expressing VGLUT1_{PM} (1548 ± 332 pA, $n = 36$), but slightly above currents in untransfected cells (22 ± 13 pA, $n = 10$).”

“3. Page 4: What is an “inward holding current” ? Please elaborate. “

The holding currents is the current at the holding potential. In the experiments described at this manuscript section we held cells at 0 mV; with internal NO_3^- , this voltage elicits an outward flux of NO_3^- , which is an inward current due to the current convention. This sentence has been removed during reorganizing the first part of the result section.

“4. What is the “late” current? Please define.”

Late currents were measured 100 ms after the voltage step. We now removed the term late and described the exact time point of current measurements whenever necessary.

“5. Why do equations 4 and 5 not contain the glutamate concentration?”

Since glutamate was held constant upon variation of pH_i , we lumped all glutamate containing terms into a constant. We now begin our analysis with a modified equation 4 that explicitly uses intra- and extracellular [glutamate].

Additional changes

1. Victor Lugo, who performed part of the new experiments of the revised manuscript, was added to the authors' list.
2. To reduce the number of supplemental figures we removed the former Supplementary Fig. 2 and the part of the result section describing these data (“Intracellular acidification was insufficient to activate VGLUT1_{PM} anion currents, but further enhanced the VGLUT1_{PM} currents activated by external acidic pH”).

3. In the original version, we fitted $E_{\text{rev}}-E_{\text{H}}$ plots for the two external [glutamate] in Fig. 3A with the same slope. We have removed this restraint in fitting and now give a separate slope for each experimental condition. In a final check, we realized that we had used different weighted linear regressions for the glutamate and aspartate data in Fig. 3a; correction of this error resulted in slight modification of the aspartate slope that did not modify any of our conclusions.
4. Because of various crucial additions in text, figures and supplementary figures that significantly increased the size of our manuscript, we had to remove the section about Molecular dynamics simulations, including the associated supplementary Fig. 6, for space restrictions.
5. We attached a zip file “Source Data.zip” containing separate MS Excel files for each figure with the source data for each panel in a separate sheet.

Reviewers' Comments:

Reviewer #1:

Remarks to the Author:

The authors have provided considerable new data to address the concerns raised. With regard to the crucial question of controls, they have now included untransfected cells in Fig. S2. However, untransfected cells are not adequate controls, particularly when studied 24 h after transfection with lipofectamine, which may itself affect membrane properties. Even empty vector would help to address the role of transfection but to support the extraordinary claims made, they must use a nonfunctional mutant. The apparent presence of currents above untransfected background in these mutants raises questions about specificity and the use of untransfected cells as controls—although the TM1 arginine is particularly important for anion transport, mutation of the TM7 arginine has abolished currents in all previous assays tested for the VGLUTs. Currents with these mutants suggest that they are not due to a VGLUT.

The authors have also attempted to address specificity in other ways. In Fig. S2, they compare the untransfected cell currents to fitted data from transfected cells, rather than to the equivalent data from transfected cells. Without a direct comparison of the equivalent data (including statistics of some sort), it is impossible to draw conclusions about significance. To corroborate the specificity of currents, the authors compare them with transporter expression. In general, these parameters correlate, consistent with a role for VGLUT2, but this is only a correlation and may reflect the extent of membrane disruption by transfection. And the third control, Rose Bengal, is both difficult to understand and problematic. Fig. S2e compares background currents at neutral external pH with activating low pH in Rose Bengal—both of these should inhibit the currents, and there is again no comparison to VGLUT-mediated currents at low pH alone. What is the purpose of comparing these two sets of putative background currents? And Rose Bengal inhibition is consistent with a role for VGLUTs, but as a noncompetitive inhibitor, there have always been questions about its specificity—hence the requirement for a nonfunctional mutant. Taken together, these alternative approaches cannot replace an appropriate transfection control.

The reason this is important is because the current paper claims very different permeation from previous work, with very little if any selectivity for large permeant anions, in contrast to the strong specificity (for glutamate even over aspartate) reported previously measuring uptake into synaptic vesicles. The authors note that even glutamate blocks the Cl currents only at high concentrations, but that is almost certainly because it was added to the non-physiological (luminal) side of the membrane—there should be much more specificity on the cytoplasmic side, and even in that study, glutamate was much more potent than aspartate, whereas in the current study, glutamate and aspartate appear indistinguishable, a result very difficult to reconcile with all of the literature. The authors cite Fig. 1 of Chang et al. as evidence for aspartate currents but review of this data shows that the VGLUT1-expressing endosomes are not clearly different from controls in terms of inward currents, which are difficult to interpret as due to aspartate since the amino acid was added to the outside of the endosomes, so should have yielded only outward currents. And in that Figure, the outward currents show no difference between controls and VGLUT1-expressing endosomes, arguing against aspartate currents. Only glutamate produces outward currents that appear different from controls, but even these overlap and the only clear finding was that outward glutamate currents are larger than aspartate currents. This report thus documented only glutamate currents due to VGLUT1, and the shift in reversal potential was due to some kind of inward currents that varied with each endosome and were hence not reliable. Any shift in E_{rev} between glutamate and aspartate is thus over-interpreting the data due to background currents not clearly related to the VGLUT.

The authors also report permeation by many inorganic anions. Although this has not been studied carefully before, the luminal activation of currents was very specific for Cl and Br, not I or NO₃. This seemed to parallel permeation, so it would greatly increase confidence in the results to determine whether activation in this assay is also specific for Cl and Br—the authors here use only activation by

external Cl. Since this study conflicts with so much previous work, it would be very helpful to determine whether halides other than Cl and Br activate the currents studied here.

With regard to the rectification, which does agree with the literature, the authors refer to inactivation of outward currents by intracellular inorganic anions and the modification of this inactivation by intracellular organic anions. This is interesting but given the uncertain signal:background, it is essential to at least touch base with a control transfected cell. In addition, the outward currents would still be carried by the entry of extracellular anion and it remains unclear whether this would be Cl or glutamate/gluconate (Fig. S5). This has some relevance for one of their most important claims, that transport of glutamate (but not the other organic anions) by VGLUT1 is coupled to H⁺ exchange.

Fig. 3a now shows the effect of internal pH on Erev, an important experiment not performed before that tests the role of pH gradient while providing allosteric activation by external protons. And it would make sense that the Cl currents would make Erev more negative than predicted by H⁺:glutamate exchange. The problem is that Erev does not seem to vary (at least in any consistent way) with addition or removal of external glutamate (Fig. 3a, black symbols). So although pH alters Erev for something in the membrane, it does not appear to be relevant for glutamate transport. And the analysis of cells dialyzed with aspartate simply suggests that some background conductance for aspartate is not sensitive to pH, not that aspartate transport by VGLUT1 differs from glutamate in coupling to H⁺. This is another potential misinterpretation of the data.

The authors address the potential confounding role of pH on Cl currents by comparing Erev at two pH with what are presumed to be equivalent Cl currents (Fig. 3b). This is a good way to normalize for the relative role of pH on Cl currents and seems to indicate that the effect of pH gradient is on the glutamate currents. It is possible that cytoplasmic pH affects the magnitude of both currents but the ratio should in principle normalize for this. This is the one piece of data that supports glutamate:H⁺ exchange but it would help to show more directly what actually happens to the magnitude of Cl currents in different cytoplasmic pH, rather than ratios of currents and Erev.

Overall, the authors have responded to all of the concerns raised but unfortunately, the data still lack a direct comparison to controls that are transfected. In addition, the lack of effect of external glutamate on Erev in Fig. 3a suggests that they are measuring Erev for something other than VGLUT1. A role for H⁺ exchange might seem very appealing and has not been addressed directly in previous work, but there are too many other observations at odds with previous work that may reflect background currents to accept this result.

Reviewer #2:

Remarks to the Author:

In this revisions, the authors have answered most of my questions. One concern remaining, which also appears to be shared by reviewer 1, is that the authors couldn't find any anions to replace glutamate, which do not cause currents. Are there really no larger organic anions that would be too large to permeate the VGLUT pore? If this is so, wouldn't that mean that any anion can dissipate the electrochemical gradient?

Reviewer 1

1. “The authors have provided considerable new data to address the concerns raised. With regard to the crucial question of controls, they have now included untransfected cells in Fig. S2. However, untransfected cells are not adequate controls, particularly when studied 24 h after transfection with lipofectamine, which may itself affect membrane properties. Even empty vector would help to address the role of transfection but to support the extraordinary claims made, they must use a nonfunctional mutant. The apparent presence of currents above untransfected background in these mutants raises questions about specificity and the use of untransfected cells as controls—although the TM1 arginine is particularly important for anion transport, mutation of the TM7 arginine has abolished currents in all previous assays tested for the VGLUTs. Currents with these mutants suggest that they are not due to a VGLUT.”

This comment is a response to a sentence in our responses that we can record currents from a neutralizing mutation of the TM7 arginine, R314A VGLUT1_{PM}, after expression in HEK293T cells and can therefore not use this mutant as negative control.

We observed dramatically reduced R314A VGLUT1_{PM} currents with internal Cl⁻ (Fig1forreviewers), however, with amplitudes above non-transfected controls and also above the currents from cells expressing a non-functional VGLUT1_{PM} mutant (see below). When using NO₃⁻ as main internal anion, we observed pH dependent currents with amplitudes close to 1 nA range at pH_o 5.5 ($I_{max} = -844.3 \pm 71.9$ pA at -160 mV, $n = 13$). They were much smaller than results in cells expressing WT VGLUT1_{PM} ($I_{max} = -4.87 \pm 3.42$ nA, $n = 18$, pH_o 5.5), but clearly above background again. R314A VGLUT1_{PM} NO₃⁻ currents resemble WT in some (voltage and pH activation, time dependence, anion selectivity), but differ in pH dependence ($pK_M = 5.2 \pm 0.2$, $n = 13$, as compared to 6.1 ± 0.02 in WT) and unitary current amplitudes obtained by noise analysis ($I_{R314A} = 26 \pm 6$ fA, $n = 13$, as compared to 39 ± 1 fA in WT with internal NO₃⁻). Taken together, these data indicate that R314A VGLUT1_{PM} is functional and thus not suited as negative control.

Fig1forreviewers. Current-voltage relationships from HEK293T cell expressing R314A VGLUT1_{PM}. Cells were perfused in standard external solution at pH 5.5 and either dialyzed with NO₃-based internal solution (means \pm CI, $N = 12$, \circ) or with Cl⁻-based solutions (individual values, $n = 5$, \blacktriangle). Current amplitudes from non-transfected cells are shown in green.

I believe that the discrepancy of our conclusion and the one from earlier studies is mainly based on two technical differences: we study currents in a larger range of voltages (up to -160 mV as compared with -120 mV) and also employ the more permeant anion NO₃⁻. Likely, our background subtraction might have improved the resolution. These dissimilarities might have allowed us to observe R314A VGLUT1_{PM} in electrophysiological recordings. In agreement with our results, the Omote group found R322A VGLUT2 to be functionally indistinguishable from WT (Juge *et al.* (2006). *J Biol Chem* **281**, 39499-39506). It is therefore not true that the

mutation of the TM7 arginine has abolished currents in all previous assays tested for the VGLUTs.

We fully agree that control experiments with non-functional transporters will be helpful to convince remaining critics. We have identified a triple mutant (H191K-H426K-D428Q) that is non-functional in every transport function, also with internal NO_3^- . As discussed with the editor in a recent zoom conference, we generated control measurements for cells transfected with this mutant and added them to Supplementary Fig. 2 (now 3). They do not show larger currents than non-transfected cells; they thus provide additional strong support that the currents we assigned to VGLUT1_{PM} are indeed mediated by this transporter and are not background currents. In addition to control experiments under standard conditions (now in Supplementary Fig. 3), we are now also showing H191K-H426K-D428Q VGLUT1_{PM} glutamate and aspartate currents at pH_i 6.5, 7.4 and 8.5 in Fig. 3a and c. These control data demonstrate that our reversal potential measurements at various pH_i are also not contaminated by endogenous currents.

Comments 2., 3. and 4. support reviewer's 1 request for transfected control that we have fulfilled in the revised version. However, we feel that the controls that are commented in these points are important, and we therefore wish to give responds also to these criticisms.

2. *The authors have also attempted to address specificity in other ways. In Fig. S2, they compare the untransfected cell currents to fitted data from transfected cells, rather than to the equivalent data from transfected cells. Without a direct comparison of the equivalent data (including statistics of some sort), it is impossible to draw conclusions about significance.*

We apologize for the lack of clarity in this figure. In the original version of Supplementary Fig. 2 we showed mean current amplitudes from transfected cells as lines. We now show means \pm CI for cells expressing WT VGLUT1_{PM}. Over a large voltage range, 95% confidence intervals of VGLUT1_{PM} currents do not overlap with confidence intervals of currents obtained with non-transfected cells and cells transfected with H191K-H426K-D428Q VGLUT1_{PM}, indicating that they are different at $p < 0.05$ over almost the whole voltage range. We furthermore provide results of Mann–Whitney U-tests at -120 mV for all anions as Supplementary Table 1. Under all tested conditions, VGLUT1_{PM} currents are significantly larger than both types of background currents.

3. *To corroborate the specificity of currents, the authors compare them with transporter expression. In general, these parameters correlate, consistent with a role for VGLUT2, but this is only a correlation and may reflect the extent of membrane disruption by transfection.*

We observed closely similar reversal potentials over a large range of expression levels (Fig. 2k), which are very negative for large polyatomic anions other than glutamate. Leaky cells do not exhibit such negative reversal potentials, and changes in current amplitudes upon variation of VGLUT1_{PM} expression can thus not be caused by membrane disruption.

Biochemical analysis of VGLUTs after expression in HEK293T cells does not provide any indication for the association of VGLUT1_{PM} with endogenous proteins that might be responsible for the currents we observe (Tan et al., 2022 *eLife* **11**, e76631, doi: 10.7554/eLife.76631). The linear correlation is thus only possible when the VGLUT1_{PM} is required/mediates the observed transport function.

4. *And the third control, Rose Bengal, is both difficult to understand and problematic. Fig. S2e compares background currents at neutral external pH with activating low pH in Rose Bengal—both of these should inhibit the currents, and there is again no comparison to VGLUT-mediated currents at low pH alone. What is the purpose of comparing these two sets of putative background currents? And Rose Bengal inhibition is consistent with a role for VGLUTs, but as a noncompetitive inhibitor, there have always been questions about its specificity—hence the requirement for a nonfunctional mutant. Taken together, these alternative approaches cannot replace an appropriate transfection control.*

We removed the earlier Fig. S2e in response to this criticism. We now moved Supplementary Fig. 2f (now 2i), in which we depict concentration-dependent changes in current amplitudes starting at 0 mM Rose Bengal, to the top of the figure. In addition, we are now showing a comparison of current amplitudes before and after Rose Bengal application with statistical test as Supplementary Fig. 2j. For every tested internal anion, RB blocks VGLUT1_{PM} currents in a highly significant manner.

Rose Bengal is one of the most potent VGLUT blocker and generally assumed to be specific. There are no reports of any endogenous currents in HEK293T cells that might be blocked by Rose Bengal. Most importantly, glutamate, aspartate and gluconate current exhibit the same pharmacology, strongly supporting the notion that they are conducted by the same transporter.

5. *The reason this is important is because the current paper claims very different permeation from previous work, with very little if any selectivity for large permeant anions, in contrast to the strong specificity (for glutamate even over aspartate) reported previously measuring uptake into synaptic vesicles.*

We demonstrate that VGLUT1 can transport glutamate as well as aspartate, however with distinct transport coupling. Whereas glutamate is transported in exchange with H⁺, aspartate is mainly transported as uniport (Fig. 3). We demonstrate that this difference in coupling is sufficient for much more effective vesicle filling with glutamate (Fig. 4).

In response to this comment, we are now additionally showing that our transporter model selectively fills vesicles with glutamate under physiological conditions (with cytoplasmic 10 mM glutamate and 10 mM aspartate) (new Fig. 4i). We can also reproduce selective glutamate uptake by VGLUTs under experimental conditions (small amounts of ³H-glutamate, without or with excess of non-labeled aspartate (Moriyama and Yamamoto (1995) *J Biol Chem* **270**, 22314-22320). This is now shown in the new Fig. 4j. We describe these new results on p. 14, line 17 - 21

A strong specificity for glutamate uptake over aspartate uptake into synaptic vesicles is exactly the result of our study. We discuss this on p.19, line 23 – 25.

6. *The authors note that even glutamate blocks the Cl currents only at high concentrations, but that is almost certainly because it was added to the non-physiological (luminal) side of the membrane—there should be much more specificity on the cytoplasmic side, and even in that study, glutamate was much more potent than aspartate, whereas in the current study, glutamate and aspartate appear indistinguishable, a result very difficult to reconcile with all of the literature. The authors cite Fig. 1 of Chang et al. as evidence for aspartate currents but review of this data shows that the VGLUT1-expressing endosomes are not clearly different from controls in terms of inward currents, which are difficult to interpret as due to aspartate since the amino acid was added to the outside of the endosomes, so should*

have yielded only outward currents. And in that Figure, the outward currents show no difference between controls and VGLUT1-expressing endosomes, arguing against aspartate currents. Only glutamate produces outward currents that appear different from controls, but even these overlap and the only clear finding was that outward glutamate currents are larger than aspartate currents. This report thus documented only glutamate currents due to VGLUT1, and the shift in reversal potential was due to some kind of inward currents that varied with each endosome and were hence not reliable. Any shift in Erev between glutamate and aspartate is thus over-interpreting the data due to background currents not clearly related to the VGLUT.

In response to this comment, we removed any discussion of this topic from the re-revised version of our manuscript.

7. The authors also report permeation by many inorganic anions. Although this has not been studied carefully before, the luminal activation of currents was very specific for Cl and Br, not I or NO₃. This seemed to parallel permeation, so it would greatly increase confidence in the results to determine whether activation in this assay is also specific for Cl and Br—the authors here use only activation by external Cl. Since this study conflicts with so much previous work, it would be very helpful to determine whether halides other than Cl and Br activate the currents studied here.

We now show activation of VGLUT1_{PM} currents by external Br⁻ and I⁻ in Supplementary Fig. 2 (for Cl⁻-based internal solutions) or in Supplementary Fig. 5d (for glutamate⁻-based internal solutions). As expected, VGLUT1_{PM} glutamate as well as Cl⁻ currents are activated by Cl⁻ and Br⁻, but not by I⁻. At 40 mM I⁻, currents are closely similar to cells perfused with solutions only containing gluconate⁻.

8. With regard to the rectification, which does agree with the literature, the authors refer to inactivation of outward currents by intracellular inorganic anions and the modification of this inactivation by intracellular organic anions. This is interesting but given the uncertain signal:background, it is essential to at least touch base with a control transfected cell. In addition, the outward currents would still be carried by the entry of extracellular anion and it remains unclear whether this would be Cl or glutamate/gluconate (Fig. S5). This has some relevance for one of their most important claims, that transport of glutamate (but not the other organic anions) by VGLUT1 is coupled to H⁺ exchange.

At symmetrical Cl⁻, VGLUT1_{PM} chloride channels are fully closed at positive potentials, and only open upon hyperpolarization (Fig. 1). Such VGLUT1_{PM} chloride channel gating would prevent Cl⁻ efflux from the synaptic vesicle, and thus be in disagreement with a presumed physiological function as Cl⁻ efflux carrier. We therefore tested how cytoplasmic glutamate – which is present in the cytoplasm of the presynaptic nerve terminal – may modify voltage-dependent gating by adding glutamate to the pipette solution.

Without internal glutamate, voltage steps to +120 mV cause deactivation and results in current activation at subsequent voltage step to -120 mV. Since currents at positive voltages are very small, we quantified the time course of deactivation by plotting current immediately after stepping to -120 mV versus the duration at the preceding +120 mV step (now Supplementary Fig. 6d). With cells dialyzed with 40 mM glutamate and 100 mM Cl⁻, such time dependent deactivation is not visible (now Supplementary Fig. 6e). These experiments demonstrate that glutamate prevents complete closure of VGLUT at positive voltages by modifying voltage-dependent “gating” processes. This effect is not unique for glutamate, representative recordings in Fig. 2 illustrate that internal dialysis with virtually all polyatomic anions permit Cl⁻ influx at

positive voltages. Since we used mixtures of glutamate and Cl^- , we assume that currents at negative voltages are carried by Cl^- and by glutamate. However, the contribution of Cl^- will be much larger because of channel-mediated transport.

In response to this criticism, we generated a revised Supplementary Fig. (now) 6, in which we additionally show averaged time courses from ten cells expressing non-functional (H191K-H426K-D428Q VGLUT1_{PM}) dialyzed with 40 mM glutamate and 100 mM Cl^- (Supplementary Fig. 6c). We did not show additional control experiments with internal Cl^- , because such results have already been provided in Supplementary Fig. 2. As under all other tested ionic conditions, H191K-H426K-D428Q VGLUT1_{PM} currents were negligible also in cells dialyzed with 40 mM glutamate and 100 mM Cl^- . In the original version, we showed currents with external glutamate or gluconate. Since we do not make any conclusion from the comparison of these two conditions, we decided to only depict results with external gluconate in the revised version.

We agree that channel-mediated Cl^- influx contributes to the reversal potentials, which we are analysing in Fig. 3. However, we performed various tests that all demonstrate that the observed shifts of E_{rev} are caused by changes in the driving force for H^+ -glutamate exchange rather than by altered open probabilities of VGLUT1 Cl^- channels.

9. *Fig. 3a now shows the effect of internal pH on E_{rev} , an important experiment not performed before that tests the role of pH gradient while providing allosteric activation by external protons. And it would make sense that the Cl^- currents would make E_{rev} more negative than predicted by H^+ :glutamate exchange. The problem is that E_{rev} does not seem to vary (at least in any consistent way) with addition or removal of external glutamate (Fig. 3a, black symbols). So although pH alters E_{rev} for something in the membrane, it does not appear to be relevant for glutamate transport.*

Reviewer 1 is wrong, the similarity of current reversal potentials with external gluconate or external glutamate does not argue against the specificity of our currents, it is rather in good agreement with our model of variable transport coupling in VGLUT; i.e. with glutamate transport coupled to H^+ exchange, but gluconate transport not. In this case, exchanging external glutamate to gluconate increases the driving force for glutamate outward transport, resulting in larger current amplitudes at negative voltages. However, gluconate inward transport is uncoupled; it occurs along a much larger driving force than glutamate inward transport that is stoichiometrically coupled to H^+ outward transport against an 100fold concentration gradient. The increase of two currents in opposite directions can compensate for each other and thus result in only small changes of the reversal potential.

To test experimentally, whether such processes apply for VGLUT1_{PM}, we determined mean current-voltage relationships from cells that are internally dialyzed with glutamate as sole anion (Supplementary Fig. 7e). Cells were externally perfused with 40 mM Cl^- and either 100 mM gluconate or glutamate, and currents were normalized to peak current amplitudes at -160 mV upon external perfusion with gluconate-based solution.

Changes from glutamate to gluconate resulted in small increases of current amplitude and slight changes in the reversal potential. For external glutamate, the data can be well fit with sums of Cl^- currents predicted by the Goldman-Hodgkin-Katz equation and coupled H^+ -glutamate exchange currents (with 1:1 stoichiometry), multiplied with a Boltzmann term to account for voltage-dependent rectification. Using the same parameters, however, with external [glutamate] reduced to 10 pM and additional passive gluconate influx, this equation describes well current-voltage relationships after substitution of external glutamate with gluconate. This quantitative description does not permit external glutamate levels of 0. However, 0 mM glutamate levels

are not likely in experiments, since VGLUT1_{PM} mediates glutamate efflux under these conditions; it thus appears fair to take a very small glutamate concentration into considerations. We tried various very low [glutamate] and could always fit the data with external gluconate well.

This analysis demonstrates that only slight changes in reversal potential upon exchanging external gluconate to glutamate are in full agreement with VGLUT1_{PM} transporting glutamate in exchange with H⁺. We now show these data as Supplementary Fig. 7e and discuss them in the result section on p. 12, line 1 – 9.

10. *And the analysis of cells dialyzed with aspartate simply suggests that some background conductance for aspartate is not sensitive to pH, not that aspartate transport by VGLUT1 differs from glutamate in coupling to H⁺. This is another potential misinterpretation of the data.*

This criticism is again based on reviewer 1's concern about possible background currents. To remove such concerns of reviewer 1 as well as of future readers we decided to show current-voltage recordings from cells transfected with WT VGLUT1_{PM} as well as with the non-functional mutant in Fig. 3, for glutamate as well as from aspartate currents. Since this point appears to be crucial for this discussion, we copied the new Fig. 3a and c as Fig2forreviewers in this rebuttle.

VGLUT1_{PM} glutamate currents shift almost parallel with increasing pH, exactly as theoretically predicted for a change in the driving force. Negative currents are increased, positive currents decreased by alkaline internal pH. In contrast, aspartate currents are only slightly affected by pH_{int}. For both, glutamate and aspartate, background currents in cells transfected with a non-functional VGLUT1_{PM} variant are negligible (shown as small symbols in grey) and not affected by internal pH. These results show that we can quantify VGLUT-mediated glutamate and aspartate currents without contamination with background currents and accurately measure current reversal potentials.

Fig2forreviewers, Late current–voltage relationships of glutamate (left) or aspartate (right) currents for various internal pH after correction for background currents. The external solution contained 40 mM Cl⁻ and 100 mM glutamate (left)/gluconate (right) and was titrated to external pH 5.5. Small symbols show background currents from cells expressing H191K-H426K-D428Q VGLUT1_{PM}.

11. *The authors address the potential confounding role of pH on Cl currents by comparing E_{rev} at two pH with what are presumed to be equivalent Cl currents (Fig. 3b). This is a good way to normalize for the relative role of pH on Cl currents and seems to indicate that the effect of pH gradient is on the glutamate currents. It is possible that cytoplasmic pH affects the magnitude of both currents but the ratio should in principle normalize for this. This is the one piece of data that supports glutamate:H⁺ exchange but it would help to show more directly what actually happens to the magnitude of Cl currents in different cytoplasmic pH, rather than ratios of currents and E_{rev}.*

We now provide supplementary Fig. 7b that shows that absolute Cl^- current amplitudes are not affected by pH_{int} .

Since there are major concerns about possible background contaminations to our currents, I would like to summarize all evidence that – in our experiments - VGLUT1 currents are not contaminated by background currents at the end of these comments. (1) We tested non-transfected HEK293T cells under all conditions and observed zero currents using the subtraction protocol for VGLUT1_{PM} currents. We furthermore tested currents in cells expressing a mutant non-functional VGLUT1_{PM}, demonstrating that such cells do not exhibit currents above background in non-transfected cells and also exhibit zero currents using the subtraction protocol for VGLUT1_{PM} currents. We additionally tested glutamate and aspartate background currents in mutant VGLUT1_{PM} expressing cells at pH_i 6.5 and 8.5. Currents are negligible also under these conditions, demonstrating that the observed pH_i dependence of VGLUT1_{PM} currents is not due to contamination by background currents, but is specific and moreover distinct for glutamate and for aspartate currents. (2) We additionally compared current amplitudes (again under all experimental conditions) for different expression levels and observed a linear relationship between number of transporter per cell and currents. Extrapolating these lines to zero expression results in very small currents that are similar to background currents in non-transfected cells. These results indicate that the currents are not due to upregulation of endogenous currents upon transfection. (3) We show that VGLUT1_{PM} currents carried by multiple anions can be blocked by Rose Bengal, thus establishing a pharmacological link to VGLUT1. Most importantly, glutamate currents are blocked with closely similar concentration dependence as aspartate and gluconate currents, strongly supporting the notion that all currents are transported by the same protein. (4) We present a mutant, H120A VGLUT1_{PM}, with clearly distinct properties under many experimental conditions. WT and H120A VGLUT1_{PM} Cl^- and NO_3^- currents exhibit distinct unitary current amplitudes and distinct time and voltage dependences. H120A VGLUT1_{PM} glutamate, aspartate and gluconate currents are similar in amplitude, but kinetically clearly distinct from WT. E_{rev} from WT, but not H120A VGLUT1_{PM} glutamate currents depend on the pH gradient across the membrane. We cannot see how these data taken together can be explained by background currents.

Reviewer 2

In this revisions, the authors have answered most of my questions. One concern remaining, which also appears to be shared by reviewer 1, is that the authors couldn't find any anions to replace glutamate, which do not cause currents. Are there really no larger organic anions that would be too large to permeate the VGLUT pore? If this is so, wouldn't that mean that any anion can dissipate the electrochemical gradient?

Reviewer 2 is right, there must exist anions that are too large to be transported by VGLUTs. However, we have thus far restricted ourselves to anions we are usually working with and that are impermeant through most anion channels and transporters. For example we have not tested 4 sulfonic-calix(n)arenes that were used to estimate the pore diameter of volume-activated anion channels to between 1.1 and 1.7 nm (Droogmans, et al. (1999). *Br J Pharmacol* **128**, 35-40, doi:10.1038/sj.bjp.0702770). We will not claim that there are no impermeant anions anywhere in the manuscript.

In synaptic vesicles, glutamate is accumulated, utilizing the proton electrochemical gradient actively generated by the V-ATPase. Our work describes how a promiscuous transporter can mediate highly selective glutamate accumulation via variable transport coupling. Since only glutamate transport is coupled to H^+ , glutamate is taken up with much higher driving force. In the absence of glutamate, other anions may be taken up, however, at much lower driving forces, which are likely not sufficient to permit efficient uptake at physiological voltages in synaptic vesicles.

To illustrate selective transport of our model more directly, we are now showing predictions of our model for vesicular glutamate uptake in the additional presence of aspartate (Fig. 4i and j). Under physiological glutamate and aspartate concentrations (Fig. 4i) as well as under radiotracerflux conditions with an excess of external aspartate (Fig. 4j), variable transport coupling of VGLUTs predicts highly selective glutamate uptake.

Reviewers' Comments:

Reviewer #1:

Remarks to the Author:

The authors have largely alleviated previous concerns about background currents contaminating the analysis. The identification of a triple mutant with no currents is an improvement over the previous use of a TM7 arginine mutant with some residual currents even if no activity with that single mutant was observed by most others. It would be nice to show that the triple mutant is in fact expressed but assuming it is, this increases confidence in the results, and it at least provides a control that still involves transfection. The analysis of Rose Bengal is also more clear, and the authors thus appear to have addressed the issue of specificity. In addition, the demonstration that Br⁻ and Cl⁻ but not I⁻ activate the currents touches base with the literature, increasing confidence in the results.

In addition to the question of specificity, the revision includes interesting new data about the role of cytoplasmic anions in voltage dependence of the currents, suggesting a mechanism that would confer the efflux of Cl⁻ from synaptic vesicles.

The authors have also addressed another important issue, the minimal change in E_{rev} with replacement of glutamate by gluconate. They make a good case for the lack of change in E_{rev}, accounting for their results. However, this limitation arises because they have not found an organic anion that fails to permeate. The arguments are satisfactory as is but would be strengthened greatly by finding an impermeant anion that could be used to substitute for glutamate, in which case E_{rev} should shift. One can understand a lack of specificity but size has got to be some constraint. And with regard to specificity, they now provide modeling to suggest that the VGLUTs can discriminate between glutamate and aspartate due strictly to differences in H⁺ coupling. I am not sure there is any other way to address this, but they should also comment on the actual gradients achieved by synaptic vesicles—only 10-20-fold, whereas coupling to H⁺ antiport confers the potential for concentration up to 1000-fold.

This work will generate major controversy but the analysis appears careful and thus worth consideration.

Reviewer #2:

Remarks to the Author:

I have no further comments regarding this second revision.

Reviewer 1

1. *“The authors have largely alleviated previous concerns about background currents contaminating the analysis. The identification of a triple mutant with no currents is an improvement over the previous use of a TM7 arginine mutant with some residual currents even if no activity with that single mutant was observed by most others. It would be nice to show that the triple mutant is in fact expressed but assuming it is, this increases confidence in the results, and it at least provides a control that still involves transfection. The analysis of Rose Bengal is also more clear, and the authors thus appear to have addressed the issue of specificity. In addition, the demonstration that Br⁻ and Cl⁻ but not I⁻ activate the currents touches base with the literature, increasing confidence in the results. “*

We are now showing a representative SDS-PAGE analysis of lysates from HEK293T cells expressing GFP-tagged WT or H191K-H426K-D428Q VGLUT1PM as Supplementary Figure 2i. This analysis shows that the triple mutant expresses well in HEK293T cells.

2. *However, this limitation arises because they have not found an organic anion that fails to permeate. The arguments are satisfactory as is but would be strengthened greatly by finding an impermeant anion that could be used to substitute for glutamate, in which case Erev should shift. One can understand a lack of specificity but size has got to be some constraint.*

In transporters, substrates are usually moved via conformational changes from one membrane side to the other. This mechanism permits translocation of large substrates, but prevents the transport of smaller ones (that bind less effectively). An extreme example are the organic anion transporters, which transport an amazing range of compounds, many of them quite large (Roth et al. (2012) *Br J Pharmacol* **165**, 1260-1287). The OATs belong to the MFS family and are thus structurally related to VGLUT.

Our paper demonstrates how a promiscuous transporters mediates the perfectly selective transport of glutamate via variable transport stoichiometry, We believe that testing six polyatomic anions is sufficient to support this mechanism. VGLUT selects without size selectivity, and even if additional anions might be transported at very negative voltages under voltage clamp, they will not be accumulated in synaptic vesicles. We feel that understanding whether and how VGLUTs further distinguish between uncoupled anions – all not transported under physiological conditions – is out of the scope of this manuscript. We will certainly address this question in the near future. We comment on this issue on p. 18 , line 6 to 17 of the revised manuscript.

3. *And with regard to specificity, they now provide modeling to suggest that the VGLUTs can discriminate between glutamate and aspartate due strictly to differences in H⁺ coupling. I am not sure there is any other way to address this, but they should also comment on the actual gradients achieved by synaptic vesicles—only 10-20-fold, whereas coupling to H⁺ antiport confers the potential for concentration up to 1000-fold.*

The H120A mutation - that abolishes H⁺ coupling of glutamate - is known to prevent radioactive glutamate uptake (Juge et al. (2006). *J Biol Chem* **281**, 39499-39506), thus providing experimental support for the postulated mechanism of specificity. We comment on this important conclusion on p. 18, line 10 to 11.

At present, most ionic conditions in synaptic vesicles as well as membrane potentials of such vesicles are not known. It is therefore not possible to predict driving forces, moreover, glutamate concentrations have not yet been measured in synaptic vesicles. We decided therefore to refrain from any discussion of concentrations gradients across the vesicular membrane.